# DyGNeX: Efficient Distributed Training of Dynamic Graph Neural Networks with Cross-Time-Window Scheduling

## Abstract

Dynamic Graph Neural Networks (DGNNs) are advanced methods for processing evolving graph data, capturing both structural and temporal dependencies efficiently. However, existing distributed DGNN training methods face challenges in achieving load balance across GPUs and minimizing communication overhead, which limits their efficiency. In this paper, we introduce DyGNeX, a distributed training system designed to address this issue. DyGNeX utilizes a cross-time-window snapshot group scheduling algorithm that balances computational loads across GPUs without introducing additional cross-GPU feature aggregation or hidden state communication. Based on the specific scenario, the scheduling algorithm is applied using greedy or Integer Linear Programming (ILP) methods, referred to as DyGNeX-G and DyGNeX-L, respectively. DyGNeX-L and DyGNeX-G achieve average reductions of 28% and 24% in per-epoch training time compared to state-of-the-art methods, maintaining load imbalance across GPUs at approximately 4% and 8%, while preserving model convergence across various DGNN models and datasets. In simulation experiments, as the number of GPUs increases, DyGNeX-G shows good scalability, efficiently handling clusters with up to 512 GPUs while maintaining 95% efficiency.

## 1 Introduction

Dynamic graphs are graphs whose structures and attributes change over time. Dynamic Graph Neural Networks (DGNNs) have emerged as state-of-the-art methods for processing dynamic graphs, exhibiting strong ability to capture both structural and temporal dependencies (Zhu et al., 2016; Zhou et al., 2018; Wu et al., 2018; Trivedi et al., 2019; Pareja et al., 2020; Manessi et al., 2020; Chen et al., 2020; Xu et al., 2020; Goyal et al., 2020; Sankar et al., 2020; Wang et al., 2021a;b; Bai et al., 2022; You et al., 2022; Wang et al., 2022; Tian et al., 2023; Li et al., 2024; Zhang et al., 2024). Depending on the event model, dynamic graphs are categorized into Discrete Time Dynamic Graphs (DTDGs) and Continuous Time Dynamic Graphs (CTDGs). DGNNs are categorized similarly according to the dynamic graphs they process. In this work, we focus on DGNNs designed for DTDGs, which process temporal dynamics in discrete snapshots.

Significant efforts have been made to improve the efficiency of DGNN training. Some works focus on efficient training on a single GPU (Li & Chen, 2021; Guan et al., 2022; Qin et al., 2023; Wang et al., 2023; Gao et al., 2024a;b; Su et al., 2024), addressing various factors such as memory footprint and data access overhead. Others consider distributed training on multiple GPUs (Chakaravarthy et al., 2021; Fu et al., 2023; Chen et al., 2023). ESDG (Chakaravarthy et al., 2021) distributes temporally adjacent snapshots across different GPUs and requires hidden state transfers between GPUs for temporal processing, which incurs a significant overhead when the hidden states are large. BLAD (Fu et al., 2023) avoids such overhead by processing each group of temporally adjacent snapshots on the same GPU while assigning different groups across GPUs. As we will show later in Figure 1, both ESDG and BLAD experience load imbalance across different GPUs, which results in inefficient resource utilization and hinders training efficiency. While DGC (Chen et al., 2023) attempts to balance the load, it relies on graph partitioning, which introduces additional communication overhead. Therefore, it requires further research to achieve **load balance** across GPUs while **minimizing inter-GPU communication** in distributed training of DGNNs.

To address this problem, we develop a distributed DGNN training system called DYGNEX, which uses a novel cross-time-window snapshot group scheduling algorithm for load balancing. DYGNEX takes advantage of the fact that snapshot groups from different time windows are treated as independent samples in DGNN training. This allows for the flexibility of combining and scheduling them in any order. We formulate an optimal scheduling problem to minimize the per-epoch training time. Depending on the scenario, DYGNEX solves the problem using Integer Linear Programming (ILP) or a greedy algorithm, and the resulting scheduling algorithms are denoted by DYGNEX-L and DYGNEX-G, respectively. Through real-world experiments and simulations, we demonstrate that DYGNEX-L and DYGNEX-G achieve average reductions of 28% and 24% in per-epoch training time compared to state-of-the-art methods. DYGNEX-L and DYGNEX-G reduce the average load imbalance ratio by 22% and 18% compared to the partition-by-snapshot-group (PSG) method. In simulation experiments, as the number of GPUs increases, DYGNEX-G shows good scalability, efficiently handling clusters with up to 512 GPUs while maintaining 95% efficiency.

## 2 PRELIMINARIES

**Dynamic Graph Neural Networks.** Dynamic Graph Neural Networks (DGNNs) are composed of multiple blocks that integrate both structural and temporal encoding mechanisms. Each block typically includes a structure encoder, which aggregates information from a node's immediate neighbors to capture its structural context, and a time encoder, which accumulates information over time to reflect temporal changes in the graph. The specific implementations of these encoders vary across different DGNN models. For example, EvolveGCN (Pareja et al., 2020) dynamically adjusts its graph convolutional network (GCN) parameters over time to accommodate the evolving nature of the graph. WD-GCN (Manessi et al., 2020) combines a GCN with a long short-term memory (LSTM) network to capture both spatial and temporal features in dynamic graphs. TGCN (Chen et al., 2020) integrates a GCN with a gated recurrent unit (GRU) to effectively capture spatial and temporal dynamics in dynamic graphs. GAT-LSTM (Wu et al., 2018) leverages a Graph Attention Network (GAT) for capturing structural information while using an LSTM to model temporal dependencies. Each of these models showcases unique approaches to integrating structural and temporal information, thereby enhancing the model's ability to learn from dynamic graph data. These four models are representative typical GNN and RNN models in DGNNs. Many subsequent models can be considered variants of these, including structural-specific models like TTGCN (Li et al., 2024) and DRAIN (Bai et al., 2022), temporal-specific models like SGNN-GR (Wang et al., 2022) and ROLAND (You et al., 2022), and comprehensive models such as Dyngraph2vec (Goyal et al., 2020) and DySAT (Sankar et al., 2020).

**Workflow of distributed training for DGNNs.** Training a DGNN on multiple GPUs requires careful management of graph data across devices. In distributed settings, the dynamic graph dataset $G = (G_1, G_2, \ldots, G_T)$ is typically partitioned across GPUs, introducing additional inter-GPU communication tasks required for accurate feature aggregation and temporal modeling. At each time step $t$, for each node $v$ in graph $G_t$, the aggregation function Aggregate$_v$ combines the node's feature $X_t(v)$ with those of its neighbors $\{X_t(u)|u \in N(v)\}$, producing a structural representation as shown in Equation 1.

$$H_t(v) = \text{Aggregate}_v(W_{\text{gnn}}, \{X_t(u)|u \in N(v)\}, X_t(v)) \tag{1}$$

When a neighboring node $u$ is located on a different GPU, inter-GPU communication becomes necessary to retrieve the neighbor's feature, introducing additional **neighbor feature communication**. This communication overhead can be significant, especially for large or densely connected graphs. On a global scale, the graph embedding $H_t$ at time $t$ aggregates information from both the node feature matrix $X_t$ and the graph structure $G_t$, as defined in Equation 2, with $t = i, \ldots, i + w$ for each time window.

$$H_t = \text{Aggregate}(W_{\text{gnn}}, X_t, G_t), \quad t = i, \ldots, i + w \tag{2}$$

This graph embedding is subsequently passed to a temporal model, such as an RNN, to capture time-dependent dynamics. As shown in Equation 3, the temporal update function combines the current graph embedding $H_t$ with the hidden state $h_{t-1}$ from the previous time step. When $h_{t-1}$ resides on a different GPU, additional **hidden state communication** is required to transfer $h_{t-1}$ across devices to maintain temporal dependency.

$$h_t = \text{TemporalUpdate}(W_{\text{rnn}}, H_t, h_{t-1}), \quad t = i, \ldots, i + w \tag{3}$$

| | Aligraph (Zhu et al., 2019) | ESDG (Chakaravarthy et al., 2021) | DGC (Chen et al., 2023) | BLAD (Fu et al., 2023) | DYGNEX |
|---|---|---|---|---|---|
| No Neighbor Feature Comm | ✗ | ✓ | ✗ | ✓ | ✓ |
| No Hidden State Comm | ✓ | ✗ | ✗ | ✓ | ✓ |
| Load Balance | ✗ | ✗ | ✓ | ✗ | ✓ |

Table 1: Comparison of existing distributed DGNN training methods across three key dimensions: neighbor feature communication, hidden state communication and load balance. Aligraph (Zhu et al., 2019) and DGC (Chen et al., 2023) use vertex-based partitioning, ESDG (Chakaravarthy et al., 2021) applies snapshot-based partitioning, and BLAD (Fu et al., 2023) employs snapshot group-based partitioning. DYGNEX achieve load balance without introducing communication overhead for large dynamic graphs.

Efficient training in this setting requires minimizing communication overhead and maintaining load balance across GPUs, as imbalances can result in performance bottlenecks. A detailed explanation of this workflow is provided in Appendix A, with Figure 6 illustrating the process.

**Dataset Partition Strategy.** In distributed training of dynamic graphs, the mainstream dataset partitioning methods include vertex-based partitioning, represented by Aligraph (Zhu et al., 2019), and snapshot-based partitioning, represented by ESDG (Chakaravarthy et al., 2021). The latest work, BLAD (Fu et al., 2023), proposes a snapshot group-based partitioning method, which effectively reduces the overall communication volume. Specifically, in one iteration, each GPU trains a complete snapshot group. Since each snapshot contains all the node information, it avoids neighbor feature communication. Additionally, because each snapshot group includes all prior information of the target snapshot, it eliminates hidden state communication. A detailed workflow of three dataset partition strategies is provided in Appendix B, with Figure 7, 8, and 9 illustrating the differences among them.

**Dilemma in Distributed Training of DGNNs.** Achieving efficient distributed training of DGNNs requires a data partitioning strategy that ensures **load balancing** across all nodes while **minimizing inter-node communication**. Existing approaches exhibit a range of strengths and weaknesses, as shown in Table 1, none of them effectively balance inter-node communication with load distribution. In different partitioning strategies, vertex-based partitioning can fine-tune the load distribution at the node level, but it typically introduces significant communication overhead. Snapshot-based and snapshot group-based partitioning use snapshots and snapshot groups as scheduling units, respectively, but both face load imbalance due to differences between snapshots. Therefore, we focus on the load imbalance issues in ESDG and BLAD. We measured task allocation across 4 GPUs during training with ESDG and BLAD on four popular datasets, using the number of nodes and edges as workload indicators. As shown in Figure 1, where the nodes and edges processed by GPU0 are used as the baseline for comparison, ESDG exhibited differences of up to 19% in node distribution and 26% in edge distribution, while BLAD showed up to 16% difference. The uneven task distribution caused GPU idling, prolonging training time and reducing efficiency, highlighting the need for better load balancing.

## 3 SYSTEM OVERVIEW

In this section, we provide an overview of the DYGNEX design by outlining our primary design objectives. Our goal is to achieve load balancing while minimizing communication overhead.

Figure 2 illustrates the comprehensive design and execution process of the DYGNEX system, which adopts a snapshot group-based dataset partitioning approach. DYGNEX first assigns each task to GPUs for training time measurement. Then, DYGNEX profiler collects and analyzes the timing data for each task, providing a fine-grained view of system performance over time. To minimize the impact of random variations in single-sample measurements, DYGNEX sampler measure the training time for each task multiple times, ensuring a more accurate and reliable performance profile. Building on the training time data of each task, we then implement a task grouping strategy using a cross-time-window group combination algorithm. This algorithm combines tasks across different time windows, achieving effective load balancing across nodes, which is critical for improving system efficiency and scalability. In the final phase, DYGNEX deploys the newly combined tasks to

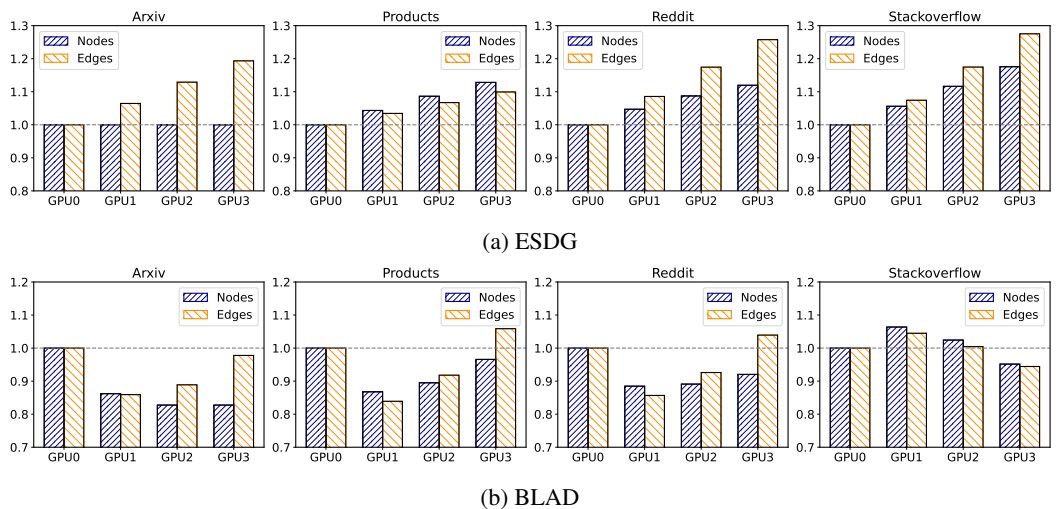

(a) ESDG

(b) BLAD

Figure 1: Load imbalance among GPUs across different datasets.

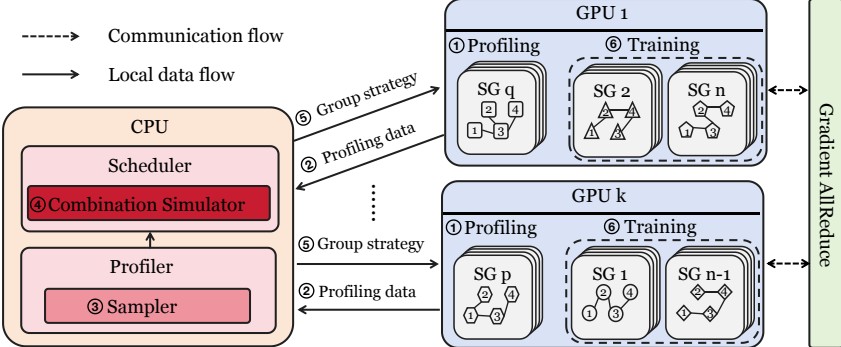

Figure 2: System Overview.

each node for training, executing the combined snapshot groups sequentially. This approach balances the training processes, maximizes node utilization, and leads to overall system performance improvements and faster convergence times.

## 4 METHOD

In this section, we introduce the two implementations of our cross-time-window task grouping algorithm: DYGNEX-L, which uses Integer Linear Programming (ILP), and DYGNEX-G, which adopts a greedy approach to reduce computational complexity. Both implementations are designed to optimize the objectives of DYGNEX by addressing the key challenges of load balancing and inter-GPU communication, ultimately improving the efficiency of distributed DGNN training. The commonly used notation in this section is summarized in Table 2.

Table 2: Frequently Used Notations

| | | | |
|---|---|---|---|
| $T$ | Total training time for one epoch | $a_i$ | Iteration assigned to snapshot group $i$ |
| $T_i$ | Time for the $i$-th iteration | $\alpha$ | Time for gradient allreduce across GPUs |
| $m$ | Number of iterations per epoch | $W_i$ | Waste time for the $i$-th iteration |
| $G$ | Number of GPUs | $g_i$ | GPU assigned to snapshot group $i$ |
| $n$ | Number of snapshot groups | $t_i$ | Execution time of snapshot group $i$ |

### 4.1 OPTIMIZATION OBJECTIVES

The primary optimization goal of DYGNEX is to minimize the total training time for one epoch, denoted as $T$. This is particularly important in distributed environments where inefficient task scheduling and uneven load distribution can result in substantial delays. The objective is mathematically formulated as:

$$\text{minimize} \quad T = \sum_{i=0}^{m} T_i, \tag{4}$$

where $m$ represents the number of iterations per epoch, and $T_i$ denotes the time for the $i$-th iteration. The duration $T_i$ of each iteration is given by:

$$T_i = \max_{1 \leq j \leq G} \left( \sum_{k=1}^{n} \mathbb{I}_{a_k=i} \cdot \mathbb{I}_{g_k=j} \cdot t_k \right) + \alpha, \tag{5}$$

where $G$ is the number of GPUs, $n$ is the number of snapshot groups, $g_k$ is the GPU assigned to snapshot group $k$, and $t_k$ is its execution time of snapshot group $k$. $a_k$ represents the iteration assigned to snapshot group $k$, and $\alpha$ is the time for gradient allreduce, which synchronizes gradient updates across GPUs. The indicator function $\mathbb{I}_{\text{condition}}$ is 1 if the condition is true, and 0 otherwise. This formulation ensures that the total execution time accounts for both task scheduling and inter-GPU communication.

Since different snapshot groups can be executed independently, determining the optimal task scheduling strategy $Strategy = [(a_1, g_1), (a_2, g_2), \ldots, (a_n, g_n)]$ that minimizes $T$ is an NP-hard problem. This is because it generalizes the classical makespan minimization problem on parallel machines, which is a well-known NP-hard problem. For a single iteration ($m = 1$) and without gradient allreduce ($\alpha = 0$), the problem reduces to assigning tasks to GPUs to minimize the maximum completion time, which is NP-hard for two or more machines. To ensure system convergence while making the solution more tractable, we impose a constraint that limits each GPU to handle at most two tasks per iteration. Then, we propose DYGNEX-L, an ILP-based approach, and DYGNEX-G, a greedy algorithm, to efficiently address this scheduling problem.

### 4.2 DYGNEX-L

To globally optimize the total training time $T$, we first propose an ILP model, DYGNEX-L. The binary decision variable $x_{k,i,j}$ indicates whether snapshot group $k$ is assigned to iteration $i$ on GPU $j$. The objective function is to minimize the total iteration time $T_i$, calculated as:

$$T_i = \max_{1 \leq j \leq G} \left( \sum_{k=1}^{n} x_{k,i,j} \cdot t_k \right) + \alpha, \tag{6}$$

The key constraints in DYGNEX-L are expressed mathematically as follows. The assignment constraint ensures that each snapshot group is assigned to exactly one GPU in one iteration:

$$\sum_{i=1}^{m} \sum_{j=1}^{G} x_{k,i,j} = 1, \quad \forall k \in \{1, 2, \ldots, n\}. \tag{7}$$

The GPU capacity constraint ensures that no GPU can process more than $L$ snapshot groups in any iteration:

$$\sum_{k=1}^{n} x_{k,i,j} \leq L, \quad \forall i \in \{1, 2, \ldots, m\}, \forall j \in \{1, 2, \ldots, G\}. \tag{8}$$

To solve this ILP problem (Equation 6), we use standard linearization techniques and off-the-shelf solvers like Gurobi (Gurobi Optimization, LLC, 2022). However, as the problem is still NP-hard, we implement a strategy that outputs the solution once it is within a specified distance (e.g., 2%) from the optimal. For further details, see Appendix C.

### 4.3 DYGNEX-G

To mitigate the computational complexity of solving the global optimization problem, we also propose DYGNEX-G, a greedy algorithm. Greedy approaches are heuristics that can efficiently solve NP-hard problems, though they often produce suboptimal solutions. DYGNEX-G attempts to reduce resource waste and balance GPU workloads by iteratively optimizing the execution sequence of snapshot groups.

The algorithm follows three main steps: (1) Generate candidate target training times $T_{target}$ by combining the longest remaining execution time with other remaining times. (2) For each $T_{target}$, select $G - 1$ tasks that minimize the deviation from $T_{target}$ using a two-pointer search strategy. (3) Compute the waste time $W_i$ for each $T_{target}$, selecting the one that minimizes $W_i$. Algorithm 1 provides an overview of the DYGNEX-G process.

---

**Algorithm 1:** DYGNEX-G Overview

---

**Input:** $G$, $res = n$, $snapshot\_groups = [t_1, t_2, \ldots, t_n]$, $iteration = 1$
**Output:** $Strategy$
Add zero group to $snapshot\_groups$ and sort it.
**while** $res > G$ **do**
    Create $target\_list$ from the longest remaining time and other remaining times;
    **for** $T_{target}$ in $target\_list$ **do**
        Find $G - 1$ pairs closest to $T_{target}$;
        Compute $W_{iteration}$;
        Update $W_{min}$ if necessary;
    Update $res$, $snapshot\_groups$, $iteration$, and record the best combination in $Strategy$;
Record remaining group in $Strategy$;
**return** $Strategy$;

---

**Group preprocessing.** Initially, the algorithm preprocesses the task list, adding a **zero group** (with execution time $t_0 = 0$) to the list to preserve flexibility in task combinations. All groups are then sorted by execution time $t_i$ to allow for efficient pairing and combination in later steps.

**Target training time selection.** Rather than directly determining the optimal $T_{target}$, the algorithm considers a range of candidate target times, starting with the group with the longest remaining time and all possible pairwise combinations. This ensures a comprehensive search without being computationally prohibitive.

**Group selection and waste time calculation.** For each candidate $T_{target}$, a two-pointer search identifies group combinations that best match $T_{target}$, minimizing the need for exhaustive searches. The waste time $W_i$, defined as:

$$W_i = (T_i - \alpha) \cdot G - \sum_{j=1}^{G} \sum_{k=1}^{n} \mathbb{I}_{a_k=i} \cdot \mathbb{I}_{g_k=j} \cdot t_k, \tag{9}$$

is used to evaluate the quality of each scheduling strategy. The algorithm selects the $T_{target}$ that results in the lowest $W_i$, ensuring efficient task scheduling.

**Complexity analysis.** The overall time complexity of DYGNEX-G is $O(n^3)$, making it computationally feasible for large-scale scenarios. For more details about DYGNEX-G, see Appendix D.

## 5 EVALUATION

In this section, we first introduce our experimental testbed, along with the models, datasets, and baselines employed in our evaluations. We then evaluate the performance of DYGNEX by examining its improvements in training throughput and ensuring that it does not degrade training accuracy, as well as the results from the simulation and end-to-end time analysis.

Table 3: Attributes of the Four Datasets. The symbols $|V|$ and $|E|$ denote the total number of nodes and edges. $\overline{|V|}$ and $\overline{|E|}$ represent the average number of nodes and edges per snapshot. The term $d_v$ represents the dimension of the node features. The parameters $\beta$ and $\gamma$ indicate the average degree and the number of snapshots, respectively.

| Dataset | $|V|$ | $|E|$ | $\overline{|V|}$ | $\overline{|E|}$ | $d_v$ | $\beta$ | $\gamma$ |
|---|---|---|---|---|---|---|---|
| Arxiv (Hu et al., 2020) | 169,343 | 2,409,625 | 169,340 | 1,317,917 | 128 | 7.8 | 30 |
| Products (Hu et al., 2020) | 286,010 | 16,567,128 | 167,570 | 7,268,265 | 100 | 43.4 | 30 |
| Reddit (Reddit, n.d.) | 80,125 | 47,804,919 | 62,590 | 22,183,258 | 602 | 354.4 | 30 |
| Stackoverflow (Stack-Overflow, 2023) | 2,601,977 | 63,497,050 | 160,877 | 1,269,941 | 50 | 7.9 | 50 |

## 5.1 METHODOLOGY

**Testbed.** We conduct our experiments using four A100 80GB SXM4 GPUs, connected via PCIe with a peak bandwidth of 32GB. The experiments are carried out within the DGL NGC Container (version 24.07-py3), which includes DGL v2.4 (Wang, 2019) for scalable graph processing, PyTorch v2.4.0 (Paszke et al., 2019) as the deep learning framework, and CUDA 12.5 for GPU acceleration, providing an optimized environment for our distributed graph training tasks.

**Datasets.** We use four dynamic graph datasets to evaluate the performance of DYGNEX. The Stackoverflow (Stack-Overflow, 2023) dataset is a real-life temporal network of interactions on the Stack Exchange website Stack Overflow. Additionally, we use three large-scale static graph datasets: Arxiv, Products (Hu et al., 2020), and Reddit (Reddit, n.d.). To simulate dynamics in these static datasets, we follow Fu et al. (2023) to create snapshots by randomly deleting some of the edges from the static graph. The time window size for all datasets is set to 4. The evolution pattern of the number of nodes or edges in these snapshots mirrors the trend observed in the Stackoverflow dataset, with specific details on the changes in nodes and edges provided in Figure 10 of the Appendix E.

**Benchmark DGNN models.** Four representative DGNNs are employed: EvolveGCN (Pareja et al., 2020), WD-GCN (Manessi et al., 2020), TGCN (Chen et al., 2020), and GAT-LSTM (Wu et al., 2018), as they are typical GNN and RNN models in DGNNs. The first three models are GCN-based DGNNs, while GAT-LSTM is a GAT-based model. These models are widely used due to their effectiveness in dynamic graph learning. Each DGNN model features a two-layer architecture, comprising a feature update operation and a graph aggregation operation. In EvolveGCN, the RNN updates the GNN parameters across snapshots, whereas in the other models, the RNN processes intermediate node features within the snapshots.

**Baselines.** We compare DYGNEX-G and DYGNEX-L with ESDG (Chakaravarthy et al., 2021), partition-by-snapshot-group (PSG) method and BLAD (Fu et al., 2023). ESDG is a widely used baseline for distributed DGNN training, while BLAD represents the current state-of-the-art (SOTA) approach. In ESDG, snapshots within a snapshot group are evenly distributed across GPUs based on their temporal intervals, as illustrated in Figure 8. In PSG, each GPU training a single snapshot group, as illustrated in Figure 9. BLAD utilizes a two-stage pipeline to collaboratively train two consecutive snapshot groups. In contrast, DyGNeX-G and DyGNeX-L execute two scheduled groups sequentially.

## 5.2 EXPERIMENTAL RESULTS

**Overall Performance.** We first compared the epoch training time, defined as the time required to train one epoch. The experimental results in Table 4 show that DYGNEX-L significantly reduces the epoch training time compared to other methods. Specifically, DYGNEX-L reduces the epoch training time by 49.5% to 91.1% over ESDG, 7.9% to 61.6% over BLAD, 3.9% to 29.7% over PSG, and 1.9% to 13.6% over DYGNEX-G. DYGNEX-L optimizes load balancing across GPUs from a global view without introducing additional communication overhead, resulting in the highest throughput performance. ESDG, on the other hand, suffers from reduced throughput due to the frequent transfer of hidden states between GPUs. While this impact is minimal for EvolveGCN, which has relatively small hidden states, the performance drops significantly for WD-GCN, TGCN, and GAT-LSTM, where the hidden states are larger. BLAD suffers in large dynamic graph scenarios primarily due to its lack of fine-grained load balancing across GPUs, which significantly

Table 4: Epoch Training Time (Seconds) for Different Methods Across Various Models and Datasets

| | Arxiv | | | | Products | | | |
|---|---|---|---|---|---|---|---|---|
| | EvolveGCN | WD-GCN | TGCN | GAT-LSTM | EvolveGCN | WD-GCN | TGCN | GAT-LSTM |
| ESDG | 1.03 | 12.44 | 9.70 | 13.57 | 2.41 | 23.23 | 19.18 | 24.04 |
| BLAD | 1.10 | 1.13 | 1.31 | N/A | 2.71 | 3.20 | 3.28 | N/A |
| PSG | 0.62 | 1.19 | 1.08 | 1.35 | 1.48 | 3.98 | 3.95 | 4.66 |
| DYGNEX-G | 0.55 | 1.12 | 1.06 | 1.26 | 1.09 | 3.04 | 3.31 | 3.95 |
| DYGNEX-L | **0.52** | **1.04** | **1.02** | **1.21** | **1.04** | **2.92** | **3.17** | **3.85** |
| | Reddit | | | | Stackoverflow | | | |
| | EvolveGCN | WD-GCN | TGCN | GAT-LSTM | EvolveGCN | WD-GCN | TGCN | GAT-LSTM |
| ESDG | 8.41 | 24.92 | 26.11 | 29.56 | 1.71 | 28.96 | 23.84 | 29.27 |
| BLAD | 7.81 | 6.59 | 6.01 | N/A | 1.99 | 4.03 | 3.83 | N/A |
| PSG | 3.88 | 7.02 | 7.22 | 7.33 | 0.55 | 4.02 | 3.91 | 4.51 |
| DYGNEX-G | 3.10 | 6.01 | 5.34 | 6.10 | 0.53 | 3.94 | 3.79 | 4.19 |
| DYGNEX-L | **2.78** | **5.58** | **5.33** | **5.27** | **0.52** | **3.86** | **3.44** | **3.77** |

impacts its overall performance. Moreover, in cases where a single snapshot group can fully utilize the computational resources, BLAD's strategy of processing multiple snapshot groups in parallel fails to achieve speedup, with performance even falling behind PSG. It is also worth noting that BLAD's current implementation is specifically optimized for models like EvolveGCN, WD-GCN, and TGCN. This requires a customized design for each DGNN model to fit within BLAD's training framework, making it unable to achieve out-of-the-box high performance for other models such as GAT-LSTM. As a result, the throughput for GAT-LSTM is not meaningful for comparison.

**Test Accuracy.** In DYGNEX-L and DYGNEX-G, the number of snapshot groups trained in a single iteration is up to twice that of PSG, which is equivalent to increasing the training batch size. To ensure that this does not lead to any accuracy degradation, we compared the test accuracy and loss over 100 epochs between DYGNEX-L, DYGNEX-G, and the PSG method on the Arxiv, Products, and Reddit datasets. The StackOverflow dataset, lacking labels, is not included in the accuracy comparison. As shown in Figure 3, the test accuracy differences among DYGNEX-L, DYGNEX-G, and PSG are within 3%, demonstrating that DYGNEX-L and DYGNEX-G have minimal impact on model accuracy. While slight differences may appear in the early stages of training, the accuracy of both methods converges over time, ultimately yielding very similar results. We also present the training loss and training accuracy in Appendix F.

**Imbalance ratio.** To validate DYGNEX's improvement in imbalance ratio, we measured the imbalance ratio performance of both DYGNEX-L and DYGNEX-G compared to the baselines.We define the imbalance ratio as the training time of the most heavily loaded GPU divided by that of the least loaded GPU. To more accurately reflect the impact of load imbalance, the training time measured excludes the synchronization waiting time for each GPU, such as the time spent waiting for the hidden state to be passed from the previous GPU in ESDG. Figure 4 presents the imbalance ratio results, revealing that ESDG, BLAD, and PSG suffer from noticeable load imbalances, with average ratios of 1.20, 1.44, and 1.26, respectively. In contrast, both DYGNEX-G and DYGNEX-L achieved much lower imbalance ratios, averaging 1.08 and 1.04, respectively, highlighting the effectiveness of our scheduling strategy in distributing the workload more evenly and improving overall system performance.

## 5.3 PROFILING AND ALGORITHM SOLVING COST

The system workflow consists of three stages: profiling, algorithm solving, and training. In the profiling stage, 2-5 epochs are typically run to filter out outliers, balancing accuracy and overhead. Experiments, shown in Figure 13 in the Appendix G, demonstrate that using one profiling epochs can result in unstable data and suboptimal combinations, while profiling more than one epoch leads to more consistent throughput. The algorithm solving stage is fast, taking less than 10ms for DYGNEX-G when the number of snapshots is in the tens. DYGNEX-L can also obtain a solution within a few seconds, with a gap of less than 2% from the optimal solution. We present the solving times of DYGNEX-G and DYGNEX-L under different numbers of snapshots and gap constraints in Table 5 and Table 6. Based on extensive experimental experience, we use DYGNEX-

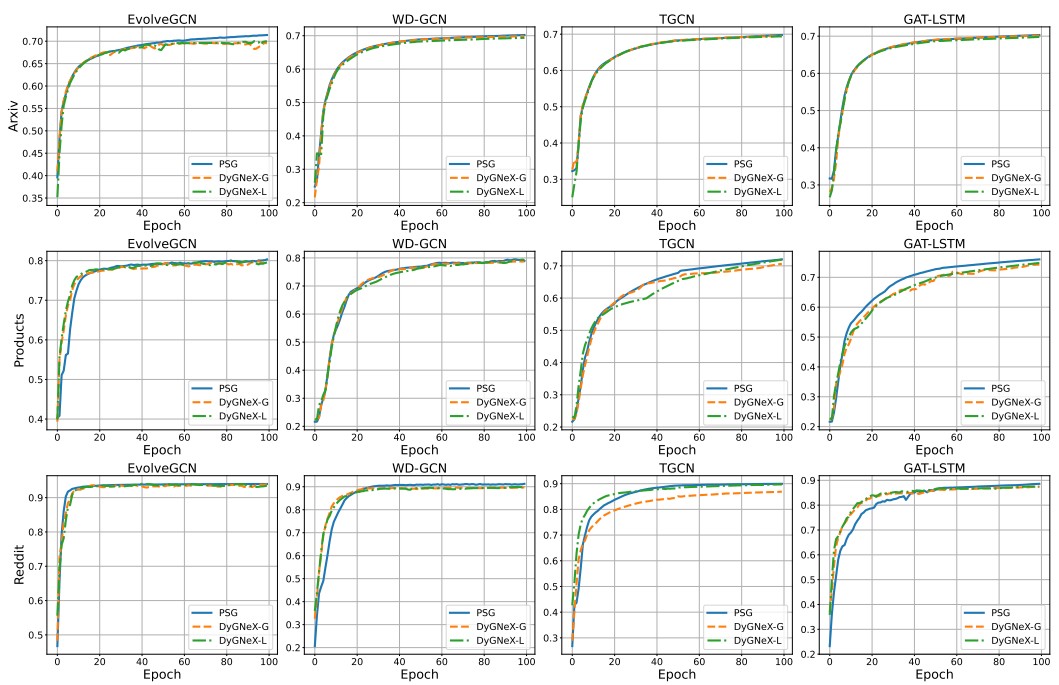

Figure 3: Test accuracy.

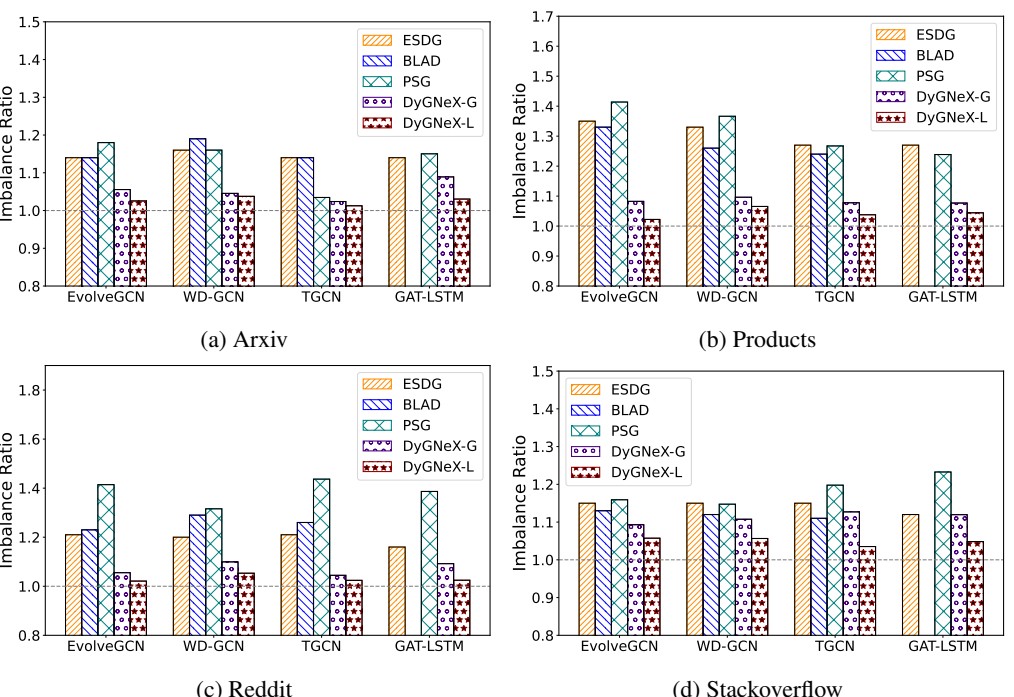

Figure 4: Imbalance ratio.

L for solving when the number of snapshots is less than 100, and DYGNEX-G when the number of snapshots exceeds 100. Finally, the training stage requires over 100 epochs for convergence, making the overhead from profiling and algorithm solving less than 3% of the total time.

## 5.4 SCALABILITY

Due to hardware limitations, we extend the evaluation to larger clusters through simulation. In this simulation, we model the dynamic graph's evolution by generating 10,000 snapshots using Dyna-Graph(Guan et al., 2022), and use a linear regression model to predict the execution time for each snapshot. The justification for the linear regression model is provided in the Appendix H. In this scenario, solving with DYGNEX-L is time-consuming, so we opt to use DYGNEX-G, which provides a faster solution, typically solving within seconds while still maintaining good performance. These predicted times are subsequently fed into DYGNEX-G for time simulation. Figure 5a shows the per-node throughput, normalized by the single-node throughput. The PSG method shows a steady drop in throughput as the number of GPUs increases, with performance degrading sharply beyond 128 GPUs. In contrast, our method scales effectively, retaining 95% efficiency at 512 GPUs and maintaining over 85% efficiency with 1024 GPUs. As shown in Figure 5b, the throughput decline is caused by the rising imbalance ratio as the number of GPUs increases. DYGNEX-G consistently maintains a lower imbalance ratio than the PSG method, which slows the performance drop.

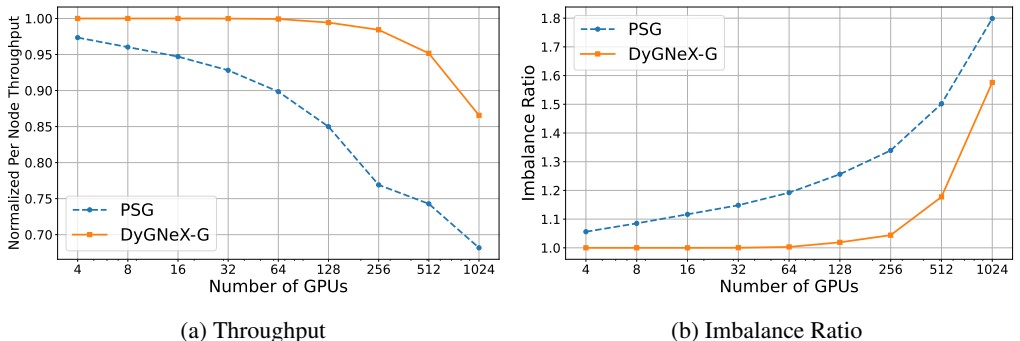

| (a) Throughput | (b) Imbalance Ratio |
|---|---|

Figure 5: Simulated per-node training throughput and imbalance ratio on clusters with 4 to 1024 nodes. Results are normalized to the throughput of training with a single node.

## 6 LIMITATIONS

**Large Snapshot Group.** In DYGNEX, each snapshot group must fit on a single GPU for training. While current GPU memory (e.g., 80GB, 40GB) suffices for most datasets, larger datasets may exceed this limit, making DYGNEX infeasible. A potential solution is to partition the graph and use full-neighbor sampling on target nodes, enabling training until all nodes are processed. The trade-offs between the overhead of partitioning and sampling and the advantages of DYGNEX over vertex-based methods (eg., DGC(Chen et al., 2023)) merit further exploration.

**Limited Number of Snapshot Groups.** The benefits of DYGNEX depend on the flexible combination of snapshot groups for load balancing. With very few groups, the limited combination space reduces potential gains.

## 7 CONCLUSION

In this paper, we introduced DYGNEX, an efficient distributed training system for DGNNs. DYGNEX addresses the challenges of load balancing and communication overhead in large-scale dynamic graph training. By utilizing a novel cross-time-window snapshot group scheduling algorithm, DYGNEX balances computational loads across GPUs without incurring additional cross-GPU communication. We implemented two variants of the system: DYGNEX-L, which uses ILP to globally optimize training efficiency, and DYGNEX-G, a greedy approach. In extensive real-world and simulated experiments, DYGNEX-L and DYGNEX-G outperform ESDG and BLAD in per-epoch training time. Both DYGNEX-L and DYGNEX-G preserve model convergence, maintaining training accuracy while improving throughput and reducing load imbalance across GPUs. DYGNEX-G further demonstrates superior scalability, efficiently handling large numbers of GPUs with minimal performance degradation.

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

## A WORKFLOW OF DISTRIBUTED TRAINING FOR DGNNs.

The entire workflow of training a DGNN model across multiple GPUs is shown in Figure 6. When graph data is partitioned across GPUs, each GPU manages a specific section of the graph, allowing for localized computations. However, since graphs are inherently interconnected, nodes often have neighbors in other partitions. This necessitates inter-GPU communication to aggregate features from neighboring nodes in different partitions, thus ensuring that each node's features include relevant information from its neighbors. This communication step, while essential for accurate feature aggregation, introduces overhead, particularly for large-scale or densely connected graphs. Communication overhead can become a significant bottleneck in the training process, as more data needs to be exchanged among GPUs, potentially slowing down the entire training pipeline. An alternative approach to mitigate this challenge is to store a complete copy of the graph on each GPU. By doing so, each GPU has access to the entire graph structure, eliminating the need for inter-GPU communication during the feature aggregation stage. This approach, however, comes with increased memory requirements, as each GPU must have enough capacity to store the full graph. For scenarios where memory is abundant and communication latency is a critical factor, this method can provide a more efficient solution. Once features are aggregated, the next phase involves processing these features through a GNN layer to generate hidden states that capture spatial dependencies within the graph structure. These hidden states are then passed to temporal models, such as RNNs or LSTMs, which capture the time-dependent dynamics inherent to dynamic graphs. The temporal model processes the sequentially evolving states to enable learning from both spatial and temporal patterns within the data.

## B DATASET PARTITION STRATEGY

The main dataset partition strategies currently include three types. First, the *vertex-based* method, as shown in Fig. 7, where each GPU's input consists of parts of multiple snapshots within a time window. Due to some nodes having neighbors on other GPUs, neighbor feature communication is required. Second, the *snapshot-based* method, shown in Fig. 8, where each GPU's input is a complete snapshot. However, since there are dependencies between snapshots within a time window, hidden state communication is needed. Finally, the *snapshot-group-based* method, shown in Fig. 9, provides each GPU with a complete snapshot group, eliminating the need for both hidden state and neighbor feature communication, making it the current state-of-the-art method.

## C INTEGER LINEAR PROGRAMMING FORMULATION

To solve the optimization problem outlined in Equation equation 4 and Equation equation 5, we develop an ILP model. This model aims to minimize the total training time $T$ for one epoch by optimally assigning snapshot groups to GPUs across iterations.

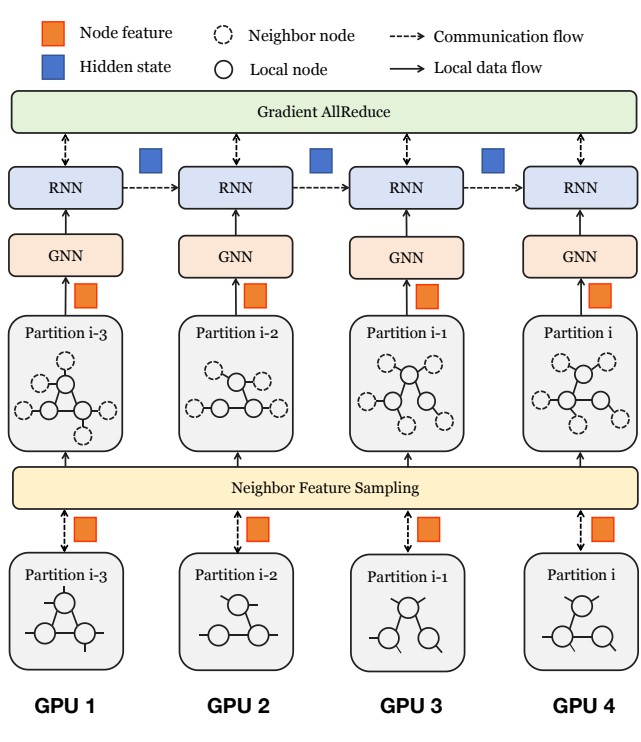

Figure 6: Workflow of distributed training for DGNNs.

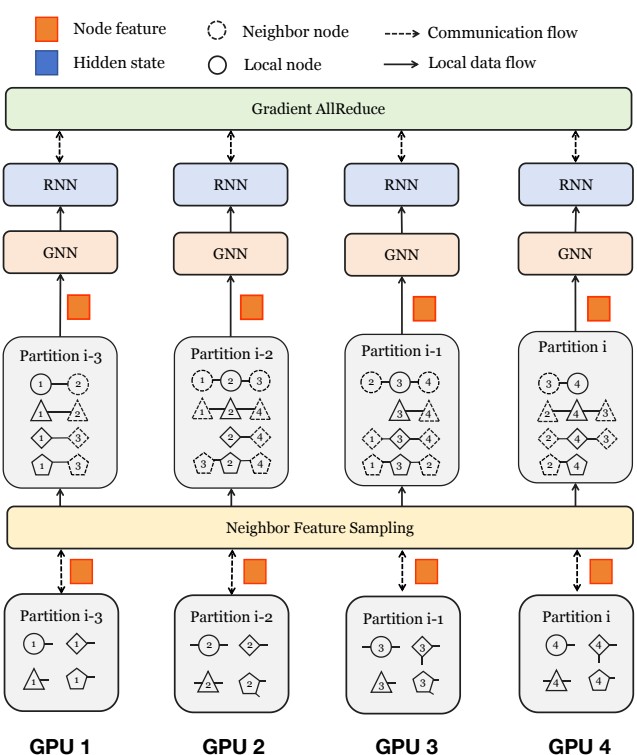

Figure 7: Workflow of vertex-based partition method.

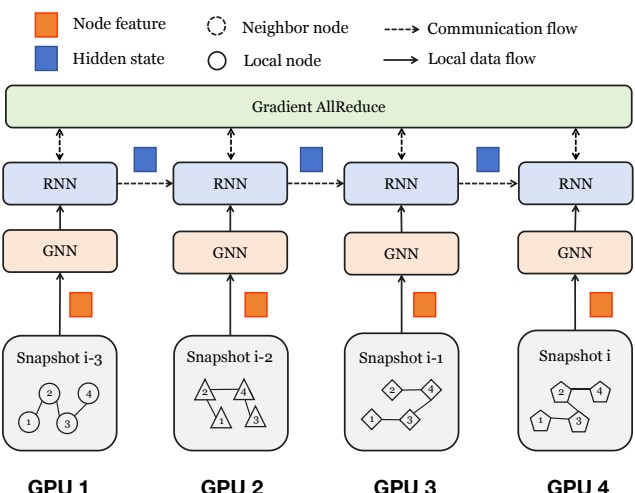

Figure 8: Workflow of snapshot-based partition method.

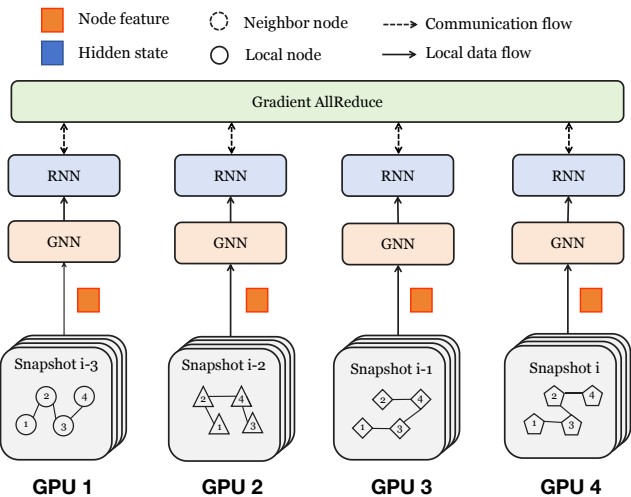

Figure 9: Workflow of snapshot-group-based partition method.

## C.1 DECISION VARIABLES

We introduce a binary decision variable $x_{k,i,j}$:

$$x_{k,i,j} = \begin{cases} 1, & \text{if snapshot group } k \text{ is assigned to iteration } i \text{ on GPU } j, \\ 0, & \text{otherwise.} \end{cases} \tag{10}$$

Here,

- $k \in \{1, 2, \ldots, n\}$ indexes the snapshot groups,
- $i \in \{1, 2, \ldots, m\}$ indexes the iterations, and
- $j \in \{1, 2, \ldots, G\}$ indexes the GPUs.

The maximum number of iterations $m$ is calculated as:

$$m = \left\lceil \frac{n}{G} \right\rceil. \tag{11}$$

## C.2 Objective Function

The objective is to minimize the total training time $T$:

$$\min T = \sum_{i=1}^{m} T_i, \tag{12}$$

where $T_i$ is the duration of iteration $i$:

$$T_i = \max_{1 \leq j \leq G} \left( \sum_{k=1}^{n} x_{k,i,j} \cdot t_k \right) + \alpha. \tag{13}$$

Here,

- $t_k$ is the execution time of snapshot group $k$,

- $\alpha$ is the time required for gradient allreduce.

## C.3 Constraints

**Assignment Constraint.** Each snapshot group must be assigned to exactly one GPU in one iteration:

$$\sum_{i=1}^{m} \sum_{j=1}^{G} x_{k,i,j} = 1, \quad \forall k \in \{1, 2, \ldots, n\}. \tag{14}$$

**GPU Capacity Constraint.** The number of snapshot groups assigned to a GPU in any iteration must not exceed a specified limit $L$ (e.g., $L = 2$):

$$\sum_{k=1}^{n} x_{k,i,j} \leq L, \quad \forall i \in \{1, 2, \ldots, m\}, \forall j \in \{1, 2, \ldots, G\}. \tag{15}$$

**Linearization of the Max Function.** To linearize the max function in the objective, we introduce auxiliary variables. Let $z_{i,j}$ be a continuous variable representing the cumulative execution time on GPU $j$ in iteration $i$:

$$z_{i,j} = \sum_{k=1}^{n} x_{k,i,j} \cdot t_k. \tag{16}$$

We introduce binary variables $u_{i,j}$ to assist in the linearization process. The following constraints ensure that $z_{i,G}$ captures the maximum execution time across all GPUs for iteration $i$:

$$z_{i,j} \geq z_{i,j-1}, \quad \forall i, \forall j \geq 2, \tag{17}$$

$$z_{i,j} \geq \sum_{k=1}^{n} x_{k,i,j} \cdot t_k, \quad \forall i, \forall j, \tag{18}$$

$$z_{i,j} \leq z_{i,j-1} + M \cdot (1 - u_{i,j}), \quad \forall i, \forall j \geq 2, \tag{19}$$

$$z_{i,j} \leq \sum_{k=1}^{n} x_{k,i,j} \cdot t_k + M \cdot u_{i,j}, \quad \forall i, \forall j, \tag{20}$$

where $M$ is a sufficiently large constant (e.g., $M = \sum_{k=1}^{n} t_k$).

## C.4 COMPLETE ILP MODEL

The complete ILP formulation is as follows:

$$\min \quad T = \sum_{i=1}^{m} z_{i,G} + m \cdot \alpha, \tag{21}$$

$$\text{s.t.} \quad \sum_{i=1}^{m} \sum_{j=1}^{G} x_{k,i,j} = 1, \quad \forall k, \tag{22}$$

$$\sum_{k=1}^{n} x_{k,i,j} \leq L, \quad \forall i, \forall j, \tag{23}$$

$$z_{i,j} \geq z_{i,j-1}, \quad \forall i, \forall j \geq 2, \tag{24}$$

$$z_{i,j} \geq \sum_{k=1}^{n} x_{k,i,j} \cdot t_k, \quad \forall i, \forall j, \tag{25}$$

$$z_{i,j} \leq z_{i,j-1} + M \cdot (1 - u_{i,j}), \quad \forall i, \forall j \geq 2, \tag{26}$$

$$z_{i,j} \leq \sum_{k=1}^{n} x_{k,i,j} \cdot t_k + M \cdot u_{i,j}, \quad \forall i, \forall j, \tag{27}$$

$$x_{k,i,j} \in \{0,1\}, \quad \forall k, \forall i, \forall j, \tag{28}$$

$$u_{i,j} \in \{0,1\}, \quad \forall i, \forall j, \tag{29}$$

$$z_{i,j} \geq 0, \quad \forall i, \forall j. \tag{30}$$

## D DYGNEX-G ALGORITHM ANALYSIS

The complexity of Algorithm 2 is primarily determined by the main while loop. The preprocessing step operates in $O(n \log n)$, which involves sorting the groups by their execution times. The GET-TARGETLIST function, which generates the list of candidate $T_{target}$ values, runs in $O(n)$, and the GETGROUPPAIR function, responsible for finding the best group combination for a given $T_{target}$, operates in $O(Gn)$. Given that the while loop executes $O(n/G)$ times, with each iteration costing $O(Gn^2)$, the overall time complexity of the algorithm is $O(n^3)$. This complexity is manageable, making the algorithm suitable for real-world applications where scalability and efficiency are crucial.

## E GROWTH TRENDS OF NODES AND EDGES IN THE DATASETS

The growth trends of nodes and edges in the different datasets are illustrated in Figure 10. In the Stackoverflow dataset, both the number of nodes and edges grow steadily over time across snapshots. This reflects the dynamic nature of the dataset, where new nodes (representing users, posts, or other entities) and edges (representing interactions or relationships) are continuously added. The trend shows a relatively consistent increase in both nodes and edges, with occasional fluctuations, indicating periods of more rapid growth in interactions compared to the addition of new entities.For the Products and Reddit datasets, the growth patterns of nodes and edges follow a similar trajectory to that of Stackoverflow, with both increasing gradually as the snapshots progress.In the Arxiv dataset, while the edge growth trend mirrors that of Stackoverflow, the number of nodes remains largely constant across snapshots. This is due to the relatively small number of nodes and edges in each snapshot, and in order to maximize the utilization of GPU resources, we avoid deleting nodes during the snapshot creation process.

## F TRAINING ACCURACY AND LOSS

Figures 11 and 12 show the training accuracy and loss curves over 100 epochs for four different DGNN models: EvolveGCN, WD-GCN, TGCN, and GAT-LSTM. The results compare the performance of PSG, DYGNEX-G, and DYGNEX-L across all models.

---

**Algorithm 2:** DYGNEX-G

---

**Input:** $G, n, t = [t_1, t_2, \ldots, t_n], i = 1$
**Output:** $S$
$t = \texttt{GroupPreProcess}(t)$
**while** $n > G$ **do**
    $target\_list = \texttt{GetTargetList}(t, n)$
    Initialize $W_i \leftarrow \infty$;
    **for** $T_{target}$ in $target\_list$ **do**
        $pair\_list = \texttt{GetGroupPair}(T_{target}[0], G, t, n)$
        $W_i = \max(\max_{j=1}^{G-1} pair\_list[j][0], T_{target}[0]) \cdot G - \sum_{j=1}^{G-1} pair\_list[j][0] - T_{target}[0]$
        **if** $W_i < W_{min}$ **then**
            $W_{min} \leftarrow W_i$;

    update $n, t, i$ ;
    Record corresponding group combination in $S$;

Record rest group in $S$;
**return** $S$;
**Function** $\texttt{GroupPreProcess}(t)$**:**
    Add zero group $t_0 = 0$ to $t$;
    Sort $t$ in ascending order;
    **return** $t$;

**Function** $\texttt{GetTargetList}(t, n)$**:**
    Initialize $target\_list \leftarrow \emptyset$;
    Add $t[n]$ to $target\_list$;
    **for** $i \leftarrow 0$ **to** $n - 1$ **do**
        $T_{pair} \leftarrow [t[n] + t[i], n, i]$;
        Add $T_{pair}$ to $target\_list$;
    **return** $target\_list$;

**Function** $\texttt{GetGroupPair}(T_{target}, G, t, n)$**:**
    Initialize $pair\_list \leftarrow \emptyset$;
    **while** $|pair\_list| < G - 1$ **do**
        Initialize $head \leftarrow 0, tail \leftarrow n$;
        Initialize $closest \leftarrow \infty, best\_pair \leftarrow \emptyset$;
        **while** $head < tail$ **do**
            $T_{\text{sum}} \leftarrow t[head] + t[tail]$;
            **if** $|T_{sum} - T_{target}| < |closest - T_{target}|$ **then**
                $closest \leftarrow T_{\text{sum}}$;
                $best\_pair \leftarrow [T_{\text{sum}}, head, tail]$;
            **if** $T_{sum} < T_{target}$ **then**
                $head \leftarrow head + 1$;
            **else**
                $tail \leftarrow tail - 1$;

        Add $best\_pair$ to $pair\_list$;
    **return** $pair\_list$;

---

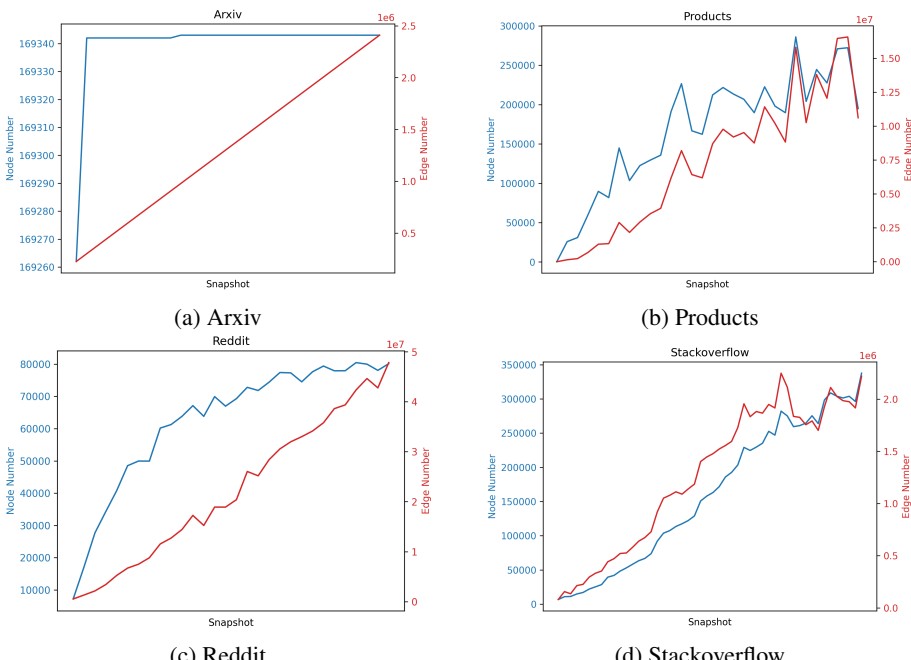

Figure 10: Growth trends of nodes and edges in the datasets

From Figure 11, it can be observed that DYGNEX-G and DYGNEX-L achieve similar accuracy to PSG throughout the entire training process. There is no significant divergence in the accuracy curves across the three methods, indicating that both DYGNEX-G and DYGNEX-L maintain comparable model performance without compromising training accuracy. This demonstrates that the scheduling techniques employed in DYGNEX-G and DYGNEX-L do not negatively affect the quality of the learned representations.Similarly, Figure 12 illustrates the training loss over time. The loss curves for DYGNEX-G and DYGNEX-L closely follow that of PSG, converging at nearly identical rates. This indicates that the optimization process is not hindered by the use of our scheduling approaches, and both DYGNEX-G and DYGNEX-L allow the models to reach the same level of loss as PSG.

## G  END-TO-END TIME ANALYSIS

The system workflow consists of three main stages: profiling, algorithm solving, and training. Typically, 3-5 epochs are run during the profiling stage to exclude outliers, and this range is chosen based on the trade-off between accuracy and overhead. We conducted experiments to analyze the impact of different profiling epoch counts on throughput, as shown in Figure 13. We found that when the profiling epoch is set to 1, the data from the first epoch is often unstable, leading to suboptimal combinations and lower throughput. When the profiling epoch exceeds 3, the profiling data becomes more stable, resulting in more consistent throughput.

The second stage is algorithm solving. When the number of snapshots is in the tens, the time consumed in the algorithm solving stage is generally less than 10ms, making it negligible.We present the solving times of DYGNEX-G and DYGNEX-L under different numbers of snapshots and gap constraints in Table 5 and Table 6. Based on extensive experimental experience, we use DYGNEX-L for solving when the number of snapshots is less than 100, and DYGNEX-G when the number of snapshots exceeds 100. Finally, the training stage requires over 100 epochs to achieve convergence. As a result, the overhead introduced by profiling and algorithm solving accounts for less than 3% of the total time. Moreover, the total time for the entire workflow in DYGNEX is significantly lower than the training time alone in both BLAD and ESDG, further highlighting the efficiency of our approach.

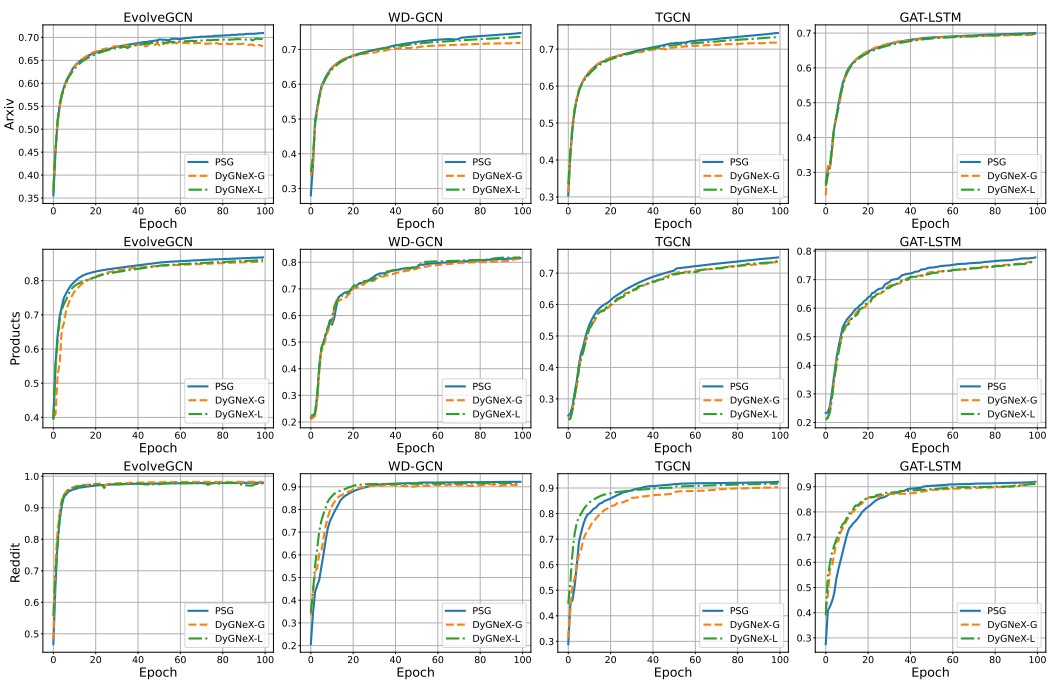

Figure 11: Training accuracy.

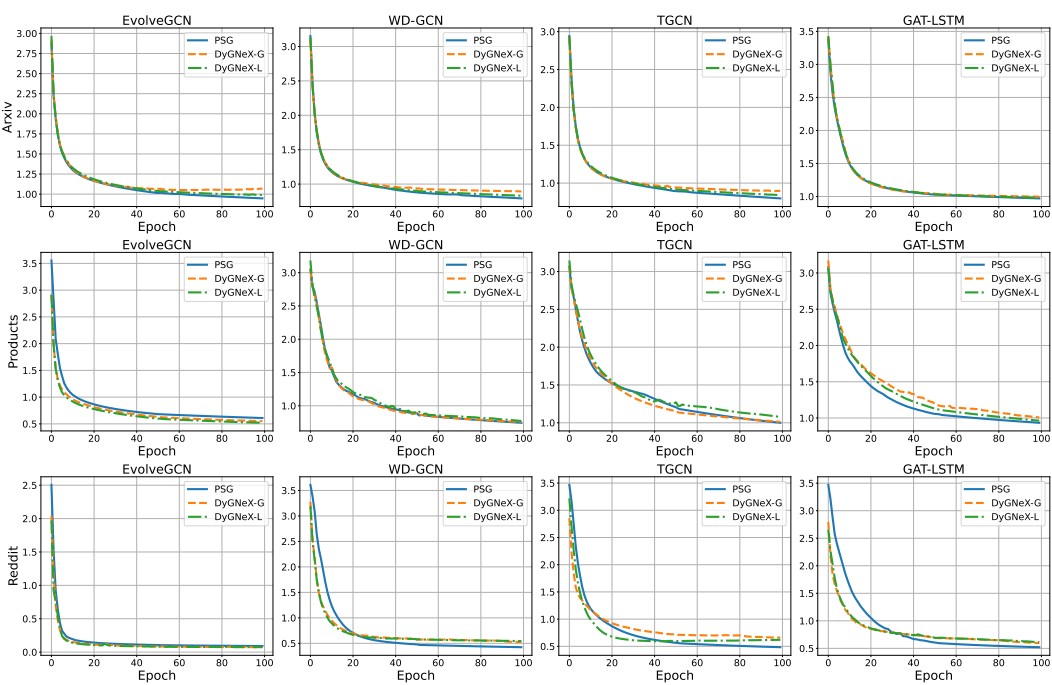

Figure 12: Training loss.

# H    SIMULATION OF SCALABILITY

In this section, we describe the logic behind our snapshot generation and time prediction. Following the approach in Dygraph(McCrabb et al., 2022), we generated 10,000 snapshots to simulate the dynamic graph evolution.

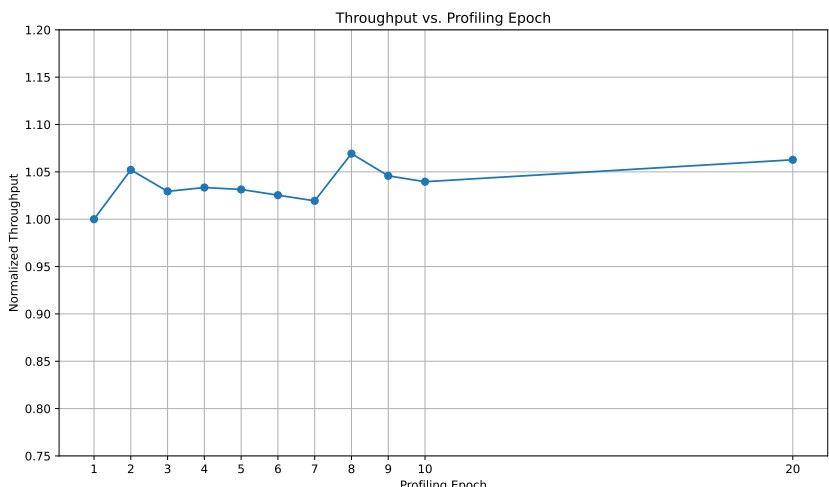

Figure 13: Throughput variation with different profiling epoch counts.

Table 5: Time cost of DYGNEX-G solving under different snapshot numbers.

| | | | | | | | | Time Cost of DYGNEX-G Solving (s) | | | | | | | | |
|---|---|---|---|---|---|---|---|---|---|---|---|---|---|---|---|---|
| | | | | | | | | Number of Snapshots | | | | | | | | |
| 10 | 20 | 30 | 40 | 50 | 60 | 70 | 80 | 90 | 100 | 150 | 300 | 500 | 1000 | 3000 | 7000 | 10000 |
| <0.001 | <0.001 | <0.001 | <0.001 | <0.001 | 0.0011 | 0.0022 | 0.0026 | 0.0021 | 0.0041 | 0.0045 | 0.0183 | 0.0395 | 0.162 | 1.297 | 6.989 | 13.880 |

Table 6: Time cost of DYGNEX-L solving under different snapshot numbers and gap constraints.

| | Time Cost of DYGNEX-L Solving (s) | | | | | | | | | | | |
|---|---|---|---|---|---|---|---|---|---|---|---|---|
| | Number of Snapshots | | | | | | | | | | | |
| Constraint | 10 | 20 | 30 | 40 | 50 | 60 | 70 | 80 | 90 | 100 | 150 | 300 |
| 1% | 0.51 | >30 | >30 | >30 | 9.22 | 11.56 | >30 | >30 | >30 | >30 | >30 | >30 |
| 2% | 0.51 | 0.60 | 1.93 | 2.63 | 3.47 | 5.36 | 4.86 | 11.16 | 18.62 | 23.02 | >30 | >30 |
| 3% | 0.48 | 0.49 | 1.09 | 1.69 | 3.07 | 5.18 | 2.68 | 4.51 | 7.92 | 9.42 | 22.08 | >30 |
| 5% | 0.46 | 0.06 | 0.22 | 0.36 | 0.71 | 0.84 | 1.02 | 1.35 | 3.63 | 1.99 | 5.00 | 25.19 |
| 6% | 0.43 | 0.06 | 0.19 | 0.23 | 0.19 | 0.81 | 1.02 | 1.05 | 2.71 | 1.98 | 3.83 | 24.73 |
| 8% | 0.43 | 0.04 | 0.14 | 0.23 | 0.19 | 0.81 | 0.40 | 1.04 | 1.53 | 1.98 | 3.82 | 17.07 |
| 10% | 0.01 | 0.03 | 0.10 | 0.23 | 0.14 | 0.35 | 0.37 | 1.05 | 1.53 | 1.96 | 3.82 | 17.05 |

For snapshot execution time prediction, we collected extensive data and profiled a subset of these snapshots on A100 GPUs, observing that the training time for a snapshot exhibits an almost linear relationship with the number of nodes and edges in the graph. We modeled the snapshot training time using the following linear equation:

$$t_{\text{snapshot}} = \alpha_1 \cdot N_{\text{node}} + \alpha_2 \cdot N_{\text{edge}} + \alpha_3 \tag{31}$$

Using another set of data for extrapolation, we found that the prediction error was less than 5%, as illustrated in Figure 14. This model was then used to simulate the execution time of each snapshot in our evaluation.

# I  MEMORY CONSUMPTION

We use `torch.cuda.max_memory_allocated()` to analyze the memory consumption during training. As shown in Table 7, we observed that DYGNEX consistently requires less GPU memory compared to BLAD across all datasets and models. The primary reason for this difference is that BLAD simultaneously launches two processes on a single GPU for data loading and training, which increases the memory requirements.

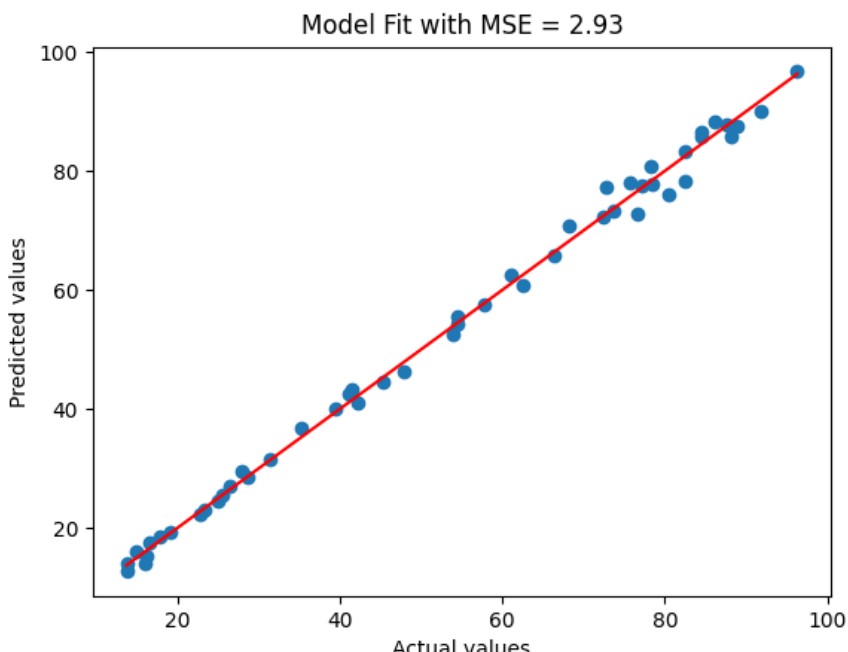

Figure 14: Comparison between predicted and measured snapshot training time.

Table 7: Memory Consumption Comparison between DYGNEX and BLAD (in GB)

| Model | Method | Arxiv | Products | Reddit | Stackoverflow |
|---|---|---|---|---|---|
| EvolveGCN | BLAD | 1.14 | 3.52 | 9.11 | 0.67 |
| | DYGNEX | 0.79 | 3.10 | 8.74 | 0.63 |
| WDGCN | BLAD | 4.29 | 14.2 | 36.07 | 5.42 |
| | DYGNEX | 0.76 | 3.09 | 8.68 | 0.63 |
| TGCN | BLAD | 1.13 | 3.47 | 9.02 | 0.70 |
| | DYGNEX | 0.77 | 2.93 | 8.68 | 0.63 |
| GAT-LSTM | BLAD | N/A | N/A | N/A | N/A |
| | DYGNEX | 0.79 | 2.93 | 8.68 | 0.63 |

