# OpenReview forum: "DyGNeX : Efficient Distributed Training of Dynamic Graph Neural Networks with Cross-Time-Window Scheduling"
_ICLR.cc/2025/Conference — ICLR 2025 Conference Withdrawn Submission_

### Official Review · Reviewer_pCCH · 2024-11-03

**Soundness:** 3
**Presentation:** 2
**Contribution:** 2
**Rating:** 3
**Confidence:** 3

**Summary:**

The paper proposes a load balancing technique for training dynamic graph neural networks on multiple GPUs. The authors evaluated their technique on a four-GPU machine and showed good scalability with simulation.

**Strengths:**

1. The proposed technique (collecting the execution time of different tasks with profiling, formalizing the task grouping problem as an optimization, and solving the optimization problem using ILP) is reasonable.

**Weaknesses:**

1. I don't see much challenge in the problem. Grouping the tasks according to the profiled execution time is straightforward.

2. Related to the first point, I don't see new insight/contribution in this paper. Load balancing has been studied extensively in many graph algorithm settings. Even for the specific task considered in this paper (distributed training of DGNNs), this paper is not the first to propose a solution.

3. While the paper targets GNNs, I don't see many ML components. The main problem it studies is the tradeoff between load balance and inter-GPU communication. This topic is more commonly studied in computer systems or high-performance computing conferences, which might be more suitable venues for the paper.

4. Evaluation with larger graphs are needed. Currently, all of the graphs used in the evaluation fit in one single A100 GPU. To show the real contribution of the proposed technique, the authors need to evaluate with graphs large enough to require distributed training.

**Questions:**

Please address my comments above.

---

> ### Author Response · Authors · 2024-11-22
>
> Thank you for your valuable review! Your insights have been instrumental in improving the quality of our paper, and we have uploaded a revised version based on your feedback. The revised sections have been highlighted in brickred for clarity.
>
> ### W1&W2. Lacks challenge & new insights or contributions
>
> Ensuring load balancing across all nodes while minimizing inter-node communication is a highly important issue in distributed DGNN training. We show that this scheduling problem is **NP-hard**. Efficiently (within seconds) obtaining approximate optimal solutions for this problem in various scenarios is challenging. DyGNeX-G and DyGNeX-L complement each other in addressing this issue under different scenarios.
>
> Although there are existing solutions in this field, we believe DyGNeX offers unique insights&contributions:
>
> 1. **Outstanding performance improvement**: DyGNeX-G and DyGNeX-L achieve an average performance improvement of 24% and 28%, respectively, compared to the state-of-the-art work BLAD.
> 2. **Complementary strategies**: DyGNeX-G and DyGNeX-L complement each other for different scenarios, providing stable and efficient solutions.
> 3. **Agnostic to DTDGs and DGNNs**: It is agnostic to the characteristics of DTDGs and DGNNs, making it highly extensible to virtually all DTDGs and DGNNs.
> 4. **Simplicity and extensibility**: It is simple, efficient, and plug-and-play, with the group strategy decoupled from the training system, enabling seamless integration with other training frameworks.
>
>
> ---
>
> ### W2. Load balancing has been studied extensively in many graph algorithm settings even for DGNN training.
>
>
> Your point is correct. There are quite a number of studies on load balancing for GNNs and DGNNs. However, the load balancing methods for traditional distributed GNN training, such as graph partitioning, cannot be directly applied to DGNN scenarios. This is because the optimization objectives in DGNN scenarios are not consistent with those of GNNs. For example, data reuse between different snapshots needs to be taken into account. Currently, the research on load balance for distributed DGNN training is insufficient. In the not-open source work DGC (SIGMOD24), it is still impossible to completely eliminate the communication tasks among GPUs other than gradient synchronization. Therefore, we believe that the topic of load balance in distributed DGNN training is well worth exploring. Meanwhile, the solution we proposed, DyGNeX, is non-trivial.
>
>
>
> ---
>
> ### W3. Focuses on system-level tradeoffs, not ML components
>
> Thank you for your suggestion; we will take it into consideration. However, the Primary Area we chose for this paper is "infrastructure, software libraries, hardware, systems, etc." DyGNeX can be viewed both as a software library that provides the group strategy and as a complete training system, which aligns well with the requirements of this Primary Area.
>
> Moreover, in ICLR 2024, there are similar works focusing on system-level contributions that have been accepted, such as:
>
> - Qi et al. (2024). *Zero Bubble (Almost) Pipeline Parallelism*. [Link](https://openreview.net/forum?id=tuzTN0eIO5)
>     - This study introduces a novel scheduling strategy that achieves zero pipeline bubbles in synchronous training by splitting the backward computation into two parts, thereby significantly enhancing the efficiency of large-scale distributed training systems.
> - Mei et al. (2024). *SRL: Scaling Distributed Reinforcement Learning to Over Ten Thousand Cores*. [Link](https://openreview.net/forum?id=lajn1iROCu)
>     - This paper presents SRL, a system designed to scale distributed reinforcement learning by splitting environment simulation, policy inference, and training into separate roles, making it easier to parallelize and balance resources across different cluster setups.
>
> Therefore, we believe that DyGNeX fits well within the area of infrastructure, software libraries, hardware, systems, etc.
>
> ---
>
> ### W4. Evaluation with larger graphs are needed
>
> The goal of DyGNeX is to leverage **data parallelism** to accelerate training, rather than to handle larger graphs. Admittedly, we consider handling larger graphs an important issue and a direction for future research, but it is beyond the scope of DyGNeX.
>
> Moreover, in most real-world scenarios, a snapshot group can easily fit into GPU memory, leaving plenty of memory to spare. DyGNeX already covers the vast majority of scenarios, and these scenarios are of significant importance.
>
> We hope our answers will address your concerns.

---

> > ### Comment · Reviewer_pCCH · 2024-11-26
> >
> > Thank you for the response. After reading your response and other reviews, my concern about limited contribution and scalability to larger graphs remains. I would keep my original score.

---

> > > ### Author Response · Authors · 2024-11-27
> > >
> > > Thank you for your prompt feedback, which has helped us gain a deeper understanding of your concerns.
> > >
> > > Regarding the issue of limited contribution, we believe that “not being the first to propose a solution” does not necessarily imply that the contribution is limited. **We think that discovering new problems is not the only way to create value**. Addressing a critical problem where existing solutions have limitations and proposing a more comprehensive solution is a highly worthwhile endeavor.
> > >
> > > As for scalability to larger graphs, **DyGNeX was not designed to handle super large graphs but to accelerate training**. Data parallel training can significantly reduce training time. At the same time, we have emphasized the broad applicability of DyGNeX. For instance, the Reddit dataset tested in Section 5 already represents a very large scenario, with a single snapshot containing over **20 million edges**. DyGNeX can run this dataset on commonly used commercial GPUs (e.g., RTX 2080 Ti, RTX 3090), let alone A100/H100 GPUs. Currently, the vast majority of dynamic graph scenarios are smaller than those discussed in Section 5. These scenarios are more practically relevant, and DyGNeX focuses on precisely these cases.
> > >
> > > We hope our answers will address your concerns.

---

### Official Review · Reviewer_V3pw · 2024-11-04

**Soundness:** 1
**Presentation:** 1
**Contribution:** 2
**Rating:** 3
**Confidence:** 3

**Summary:**

This paper introduces DyGNeX, a multi-GPU distributed training system for dynamic graph neural networks (DGNNs) designed to handle discrete-time dynamic graphs (DTDGs). DyGNeX employs a cross-time-window snapshot group scheduling algorithm to balance computational workloads across GPUs during training by distributing mini-batches and sub-mini-batches. The paper presents two variants of this algorithm, using either greedy or integer linear programming (ILP) methods for scheduling. Experimental results demonstrate promising speedup compared to the state-of-the-art approach.

**Strengths:**

S1: This paper presents an effective scheduling algorithm that addresses a crucial load balancing issue across GPUs for each training epoch—a problem previously overlooked in the DGNN literature.

S2: The proposed method is well-motivated, and its effectiveness is compelling.

S3: Experimental results demonstrate promising speedup compared to the state-of-the-art approach, BLAD.

**Weaknesses:**

W1: This paper is not self-contained. It fails to clearly and comprehensively define and discuss the research problem, technical challenges, and the proposed method.
1. Section 2 includes multiple concepts but without any figure illustrations or equations. Without reading the referenced papers, readers cannot know what the standard distributed DGNN training pipeline is, what are the current challenges, and what the contribution of this paper is. The authors should also clearly state what the dimensions in Table 1 specifically refer to (there’s a Typo in Table 1, where the caption says there are four dimensions, but the table only shows three).
2. Section 3 presents an overview of the entire system with multiple components, but the entire paper only gets into detail of the group strategy. What is the difference between other components with previous work BLAD?
3. Section 4.3 is hard to understand where a lot of symbols are not defined or explained in advance.

W2: One critical technical issue is that the proposed scheduling algorithm limits the randomness of training batches in each epoch. However, this paper does not provide rigorous theoretical analysis and requires more empirical evidence to justify how it ensures training quality. Figure 3, Figure 7, and Figure 8 show some extent of performance drop when training some DGNNs. The paper should also show how the scheduling algorithm affects training on other datasets.

W3: This paper evaluates the proposed method by comparing it with one SOTA baseline (BLAD) and another weak baseline (ESDG). The paper also introduces a strong method called PSG, which represents a naive solution with uncompromised accuracy. However, PSG lacks a detailed definition, and the differences between PSG, NyGNeX, and BLAD are not clearly stated. Clearly discussing these differences would help more accurately identify which components of the system are effective.

**Questions:**

Please see the weaknesses.

---

> ### Author Response · Authors · 2024-11-22
>
> Thank you for your valuable review! Your insights have been instrumental in improving the quality of our paper, and we have uploaded a revised version based on your feedback. The revised sections have been highlighted in brickred for clarity.
>
> ### W1.1 Lack of clarity and visual aids in Section 2
>
> Based on your suggestions, we have made the following revisions to the paper. In Section 2, we revised the description of the workflow of distributed training for DGNNs using equations and explained the communication types in **Table 1** along with their underlying reasons. Additionally, we added **Figure 6** in the appendix to illustrate the entire training pipeline. Regarding the dataset partition strategy, we included **Figures 7, 8, and 9** in the appendix to visually highlight the differences between the three strategies. Furthermore, in the “Dilemma in Distributed Training of DGNNs” section, we emphasized the current challenge: ensuring **load balancing** across all nodes while **minimizing inter-node communication**. Lastly, our contributions are clearly detailed in the final paragraph of the introduction.
>
> ---
>
> ### W1.2 The difference between other components and previous work BLAD
>
> Our system comprises three main components: the profiler, group scheduler, and trainer. The profiler in DyGNeX aligns closely with BLAD, as the profiling methodologies are largely consistent across most works. The group scheduler is the core component enabling DyGNeX to achieve load balancing by computing an optimized group strategy. This component sets DyGNeX apart from other DGNN training systems. The trainer in DyGNeX also diverges from BLAD. In BLAD, each GPU is assigned two processes, which collaboratively train different snapshots within the same snapshot group. One process handles computations for the first half of the snapshot group, while the other processes the second half, forming a two-stage pipeline. By contrast, DyGNeX executes two snapshot groups sequentially within the same process, streamlining execution as detailed in line 196.
>
> ---
>
> ### W1.3 Section 4.3 is hard to understand where a lot of symbols are not defined or explained in advance
>
> We added **Table 2** in Section 4 to explain the frequently used notations for better reader understanding. Additionally, we modified some variable names to enhance readability.
>
> ---
>
> ### W2. How the scheduling algorithm affects training on other datasets
>
> We present the test accuracy on the Products and Reddit datasets in **Figure 3**, along with the training accuracy and loss in **Figures 11 and 12**.It can be observed that in the vast majority of scenarios, there is no difference in accuracy between DyGNeX and PSG. This indicates that DyGNeX’s approach does not have a significant impact on training quality.
>
> ---
>
> ### W3. The need for a clearer definition of PSG and a more detailed comparison of the differences between PSG, DyGNeX, and BLAD
>
> Based on your suggestion, we have provided a more detailed explanation of the differences between PSG, BLAD, and DyGNeX in the baseline part of **Section 5**.
>
> In PSG, each GPU training a single snapshot group. BLAD utilizes a two-stage pipeline to collaboratively train two consecutive snapshot groups. In contrast, DyGNeX-G and DyGNeX-L execute two scheduled groups sequentially.
>
> We hope our answers will address your concerns.

---

> > ### Comment · Reviewer_V3pw · 2024-11-26
> >
> > Thank you for the detailed response and clarifications. I now understand this work better. However, due to concerns about insufficient contribution, I may not be able to change my scores.
> >
> > Based on the paper and the authors' response, PSG appears to be a variant of BLAD. DyGNeX-L demonstrates only modest improvements compared to PSG and BLAD's best results, achieving approximately 10% or less improvements in half of the settings shown in Table 4.
> >
> > This work has limited contributions in two key areas. First, from a system acceleration perspective, the group scheduler is the only optimization of DyGNeX over BLAD. Second, from a machine learning perspective, it requires theoretical analysis since the grouping strategy affects training dynamics. Given these limitations in both aspects, I decided to maintain my original score.

---

> > > ### Author Response · Authors · 2024-11-27
> > >
> > > Thank you for carefully reviewing our comments and providing a reply. We now have a deeper understanding of your concerns.
> > >
> > > **Q1. Modest improvements**
> > >
> > > **PSG can be considered an ablation baseline for DyGNeX**, and comparing DyGNeX with PSG clearly demonstrates the benefits of group scheduling. We compared the best results from DyGNeX-L, PSG, and BLAD, and found that DyGNeX-L achieved a performance improvement of 3.5% to 42.3% (**average 17%**) over max(PSG, BLAD). We believe this improvement is substantial.
> > >
> > > **Q2. Limited contributions from a system acceleration perspective**
> > >
> > > DyGNeX only inherits the snapshot group-based partitioning design from BLAD, while all other design are fundamentally different. The main design points of BLAD include:(1) operator overlap pipeline, and (2) adaptive load balance scheduling.
> > >
> > > **For the first design point**, BLAD leverages a two-level pipeline for parallel training. In scenarios involving small graphs, where memory and computation resources are abundant, BLAD achieves improvements by trading resources for time. However, in DyGNeX, to address the tight constraints on computation and memory resources in large graph scenarios and to precisely control the execution time of each group, groups in each iteration are executed sequentially rather than in a pipeline manner. **For the second design point**, BLAD only applies load balance scheduling to the two-level pipeline on the same GPU, neglecting load balance across GPUs. This is a critical aspect. DyGNeX leverages the independence among groups to perform cross-time-window task combinations, thereby improving load balancing across GPUs.
> > >
> > > It is worth noting that DyGNeX offers multifaceted improvements over BLAD in large graph scenarios, including an **average 28% throughput improvement**, **reduced memory usage** (as shown in the table below), **faster training convergence** (comparing Figure 14 in the BLAD paper with Figure 11 in this paper), and **better support for different DGNN models** (BLAD’s inter-process communication module requires customization for different DGNN models).
> > >
> > > Thus, we believe the group scheduler is not the only optimization of DyGNeX over BLAD.
> > >
> > > #### Memory Consumption Comparison between DyGNeX and BLAD (in GB)
> > >
> > > | **Model**        | **Method** | **Arxiv** | **Products** | **Reddit** | **Stackoverflow** |
> > > |-------------------|------------|-----------|--------------|------------|-------------------|
> > > | **EvolveGCN**     | BLAD       | 1.14      | 3.52         | 9.11       | 0.67              |
> > > |                   | DyGNeX     | 0.79      | 3.10         | 8.74       | 0.63              |
> > > | **WDGCN**         | BLAD       | 4.29      | 14.2         | 36.07      | 5.42              |
> > > |                   | DyGNeX     | 0.76      | 3.09         | 8.68       | 0.63              |
> > > | **TGCN**          | BLAD       | 1.13      | 3.47         | 9.02       | 0.70              |
> > > |                   | DyGNeX     | 0.77      | 2.93         | 8.68       | 0.63              |
> > > | **GAT-LSTM**      | BLAD       | N/A       | N/A          | N/A        | N/A               |
> > > |                   | DyGNeX     | 0.79      | 2.93         | 8.68       | 0.63              |
> > >
> > > **Q3. Limited contributions from a machine learning perspective**
> > >
> > > Theoretical analysis of DyGNeX is an interesting aspect, and we will place greater emphasis on theoretical analysis in our future work. However, in DyGNeX, we believe the impact is minimal. For instance, assuming a scenario with 4 GPUs and 3 iterations, our grouping strategy could be as follows:
> > >
> > > $\text{strategy}= [
> > > [g_1, g_2, g_3, g_4],
> > > [g_5, g_6, g_7, g_8],
> > > [g_9, g_{10}, g_{11}, g_{12}]
> > > ]$
> > >
> > > where $\text{strategy}[i][j] = g_k$ represents group $g_k$ scheduled on GPU $j$ during iteration $i$.
> > >
> > > In the next epoch, the strategy can undergo a certain degree of shuffling, for example:
> > >
> > > $\text{strategy} = [[g_6, g_7, g_8, g_5],[g_{10}, g_{11}, g_{12}, g_9],[g_3, g_1, g_2, g_4]]$
> > >
> > > This does not affect load balance within each iteration.
> > >
> > > It is worth noting that the mainstream approach for DGNN training currently assigns **consecutive snapshot groups sequentially** to GPUs for training.  Since consecutive snapshot groups contain overlapping snapshots (e.g., $group_1$ contains $[snapshot_1, snapshot_2, snapshot_3, snapshot_4]$, and $group_2$ contains $[snapshot_2, snapshot_3, snapshot_4, snapshot_5]$), data reuse can improve training efficiency, as demonstrated in methods like PiPAD (PPoPP’23).  However, this training approach **inherently lacks training dynamics** because the groups cannot be shuffled and must be processed sequentially.  In contrast, **DyGNeX introduces more training dynamics through out-of-order execution of groups, instead of affecting the training dynamic**. Moreover, through Figure 3, Figure 11, and Figure 12, we have sufficiently demonstrated that DyGNeX does not have a significant impact on training quality.
> > >
> > > We hope our answers will address your concerns.

---

> > > > ### Comment · Reviewer_V3pw · 2024-12-02
> > > >
> > > > I thank the authors for their detailed responses to all the concerns I raised. While their responses helped me understand this work better, they also revealed its limited contribution from both system and ML perspectives.
> > > >
> > > > I understand that incorporating randomness may help with DGNN training, and the proposed grouping strategy can balance randomness with better load distribution. This would be sufficient if the work were positioned primarily as a system acceleration study. However, the work lacks remarkable contribution from the acceleration perspective as well. The two-level pipeline for parallel training in BLAD represents a specialized optimization over sequential execution on small datasets—therefore, implementing sequential execution in DyGNeX cannot be considered an improvement over BLAD. Overall, the proposed system optimizations do not meet ICLR's standards, and this concern remained unaddressed throughout our discussion.
> > > >
> > > > Therefore, I decide to maintain my score. I sincerely hope the authors can better position and improve this work based on all the reviews.

---

> > > > > ### Author Response · Authors · 2024-12-02
> > > > >
> > > > > Thank you for your suggestions on our work. We will seriously consider your comments and improve our work. We would also like to take this opportunity to clarify certain points.
> > > > >
> > > > > ### 1. This would be sufficient if the work were positioned primarily as a system acceleration study.
> > > > >
> > > > > As claimed in Lines 61–65 of the paper, our contribution is focused on **system acceleration**, not on machine learning.
> > > > >
> > > > > ### 2. Sequential execution in DyGNeX cannot be considered an improvement over BLAD
> > > > >
> > > > > We would like to clarify that the first point in the response is simply to point out a difference between BLAD and DyGNeX. **We did not intend to claim sequential computation as a contribution or something novel**. What we would like to point out is that the two-level pipeline is a major design point (and contribution) in BLAD, and it turns out to be effective for only small graphs. DyGNeX thus got rid of it, and instead uses **cross-window scheduling for acceleration** in large graph scenario.

---

### Official Review · Reviewer_zJDK · 2024-11-04

**Soundness:** 3
**Presentation:** 3
**Contribution:** 2
**Rating:** 5
**Confidence:** 2

**Summary:**

This paper introduces DyGNex, a distributed training system for Dynamic GNN (DGNN) that balances the workload across GPUs while minimizing the inter-GPU communication. It proposes a cross-time-window snapshot group scheduling algorithm for load balances. DyGNex profiles the timing for each task on GPUs, and use a CPU to schedule using a cross-time-window group combination, which combines tasks across different time windows. To find the optimal scheduling, it uses two methods: 1) ILP; 2) greedy algorithm. The evaluation shows that DyGNex reduces the epoch training time by 2x.

**Strengths:**

* Discuss the difference between prior works (ESDG, BLAD) with DyGNex in Table 1, and Figure 1 shows the motivation (load imbalance) clearly.
* The results show that it reduces the training time per epoch and reduce the imbalance ratio.

**Weaknesses:**

* Profiling at runtime to estimate the time spent on training snapshot groups is time consuming.
* Use ILP at runtime to solve the optimal scheduling problem is also time consuming. What if it takes too long and no solution is given. How would the system proceed in such a case.
* The general idea of using runtime profiling data from different GPUs and using heuristics to schedule is not a new idea. It has been widely used in all distributed systems, like federated learning, LLM training stragglers, etc.

**Questions:**

* In line 298, what is n in the time complexity? Is it the number of snapshots for number of GPUs?
* How frequently do you need to do profiling and algorithm solving during the overall training process?
* Do you think there may be other metrics (some features in the snapshots) to estimate the runtime on each snapshot rather than doing runtime profiling?
* Table 3 shows that DyGNex reduces the training time per epoch by more than 2x, but in Figure 4, DyGNex  reduces the imbalance ratio from 1.5x to 1.0x, which is less than 2x. Could you explain why

---

> ### Author Response · Authors · 2024-11-18
>
> Thank you for your valuable review! Your insights have been instrumental in improving the quality of our paper, and we have uploaded a revised version based on your feedback. The revised sections have been highlighted in brickred for clarity.
>
> ### W1&Q2. Profiling cost
>
> In actual training, the first few epochs can be used for profiling. We only need to adjust the group strategy after the profiling epochs. The training results from the profiling epochs are not wasted, so we consider this to be a negligible overhead.
>
> We also present the solving times of DyGNeX-L under different numbers of snapshots and gap constraints in Table 1. Based on the result and our experience, we believe that when the number of snapshots is less than 100, DyGNeX-L can solve most cases within 10 seconds under a 3% constraint, which we consider acceptable. In DTDG training, the dataset is represented as $G = (G_1, G_2, \dots, G_T)$, where the dynamic connectivity of the graph **remains unchanged between epochs**. Each epoch trains on the full sequence of graph changes, $G = (G_1, G_2, \dots, G_T)$. Therefore, **profiling and solving need to be done only once**, regardless of the number of epochs, and the resulting strategy can be applied to the entire training process.
>
> #### Table 1. Time cost of DyGNeX-L solving under different snapshot numbers and gap constraints
>
> | **Constraint** | **10** | **20**  | **30**  | **40**  | **50**  | **60**  | **70**  | **80**  | **90**  | **100** | **150** | **300** |
> |----------------|--------|---------|---------|---------|---------|---------|---------|---------|---------|---------|---------|---------|
> | **1%**        | 0.51   | >30     | >30     | >30     | 9.22    | 11.56   | >30     | >30     | >30     | >30     | >30     | >30     |
> | **2%**        | 0.51   | 0.60    | 1.93    | 2.63    | 3.47    | 5.36    | 4.86    | 11.16   | 18.62   | 23.02   | >30     | >30     |
> | **3%**        | 0.48   | 0.49    | 1.09    | 1.69    | 3.07    | 5.18    | 2.68    | 4.51    | 7.92    | 9.42    | 22.08   | >30     |
> | **5%**        | 0.46   | 0.06    | 0.22    | 0.36    | 0.71    | 0.84    | 1.02    | 1.35    | 3.63    | 1.99    | 5.00    | 25.19   |
> | **6%**        | 0.43   | 0.06    | 0.19    | 0.23    | 0.19    | 0.81    | 1.02    | 1.05    | 2.71    | 1.98    | 3.83    | 24.73   |
> | **8%**        | 0.43   | 0.04    | 0.14    | 0.23    | 0.19    | 0.81    | 0.40    | 1.04    | 1.53    | 1.98    | 3.82    | 17.07   |
> | **10%**       | 0.01   | 0.03    | 0.10    | 0.23    | 0.14    | 0.35    | 0.37    | 1.05    | 1.53    | 1.96    | 3.82    | 17.05   |
>
> ---
>
> ### W2. Handling situations where solving the ILP at runtime takes too long or fails
>
> We can either relax DyGNeX-L’s constraint or switch to using DyGNeX-G.
>
> ---
>
> ### Q1. What is $n$ in the time complexity?
>
> $n$ represents the number of snapshots in the dataset. In Table 2, we show the solving time of DyGNeX-G as $n$ varies. We believe that when $n$ is less than 10,000, the solving time of DyGNeX-G is guaranteed to be acceptable.
>
> #### Table 2. Time cost of DyGNeX-G solving under different snapshot numbers
>
> | **Number of Snapshots** | 10       | 20       | 30       | 40       | 50       | 60       | 70       | 80       | 90       | 100      | 150      | 300      | 500      | 1000     | 3000    | 7000    | 10000   |
> |--------------------------|----------|----------|----------|----------|----------|----------|----------|----------|----------|----------|----------|----------|----------|----------|---------|---------|---------|
> | **Time (s)**            | <0.001   | <0.001   | <0.001   | <0.001   | <0.001   | 0.0011   | 0.0022   | 0.0026   | 0.0021   | 0.0041   | 0.0045   | 0.0183   | 0.0395   | 0.162    | 1.297   | 6.989   | 13.880  |
>
> ---
>
> ### Q3. Alternative metrics or features to estimate runtime without profiling
>
> We fully agree with your perspective, and this is indeed a direction for our future work. However, in DyGNeX, since profiling and algorithm solving only need to be performed once, the overhead is minimal, accounting for less than 3% of the end-to-end time, which we consider acceptable.
>
> ---
>
> ### Q4. Why DyGNeX achieves more than 2x reduction in training time per epoch while the imbalance ratio improvement is less than 2x
>
> DyGNeX’s performance gains stem from multiple factors, with load balancing being one of them. Compared to ESDG, DyGNeX achieves gains by both eliminating hidden state communication and enhancing load balancing. Against BLAD, DyGNeX benefits not only from superior load balancing but also from avoiding performance issues inherent to BLAD’s two-process-per-GPU setup, which introduces inter-process synchronization overhead and competition for compute resources. Relative to PSG, DyGNeX’s primary advantage lies in improved load balancing, with additional gains from reducing gradient communication frequency.

---

> > ### Comment · Reviewer_zJDK · 2024-11-24
> >
> > Thank you for the detailed response and clarifications. I appreciate the effort you've put into addressing my concerns. I understand the performance improvements you've demonstrated; however, due to concerns regarding the level of novelty, I may not be able to change my original scores.

---

> > > ### Author Response · Authors · 2024-11-27
> > >
> > > Thank you for your feedback, which has allowed us to gain a deeper understanding of your concerns.
> > >
> > > ### W. The general idea is not new & level of novelty
> > >
> > > We acknowledge your point: The general idea of using runtime profiling data from different GPUs and using heuristics to schedule is not a new idea.
> > >
> > > While both approaches involve profiling and scheduling, different problem formulations require different solutions. Our time modeling for DGNN distributed training accurately captures the critical issue of load balancing and show that the problem is **NP-hard**, which we consider to be non-trivial. We design complementary solutions(DyGNeX-G & DyGNeX-L) tailored to the specific characteristics of different scenarios, which we believe is novel.
> > >
> > > DyGNeX-L offers higher load balancing precision, while DyGNeX-G achieves second-level solving in scenarios with a large number of snapshot groups. The design of **zero groups** and the **two-pointer solving** method in DyGNeX-G provides more flexible options and greater efficiency, which we consider to be non-trivial.
> > >
> > > We hope our answers will address your concerns.

---

### Official Review · Reviewer_ryh2 · 2024-11-04

**Soundness:** 3
**Presentation:** 3
**Contribution:** 2
**Rating:** 3
**Confidence:** 2

**Summary:**

The authors introduce DYGNEX, a distributed graph neural network (GNN) training system designed to address the issue of  inefficient resource utilization across GPUs.
DYGNEX utilizes a cross-timewindow snapshot group scheduling algorithm that balances computational loads.
They integrate the scheduling algorithm into DYGNEX based on the specific scenario,  using greedy or Integer Linear Programming (ILP) methods.
From the evaluation, DYGNEX achieves an impressive performance in per-epoch training compared to ESDG, BLAD, and PSG, while preserving model convergence across various DGNN models and datasets.
In conclusion, the paper presents a novel distributed dynamic GNN training system, that could help us better understand the challenges coming from load balance across GPUs and inter-GPU communication.

**Strengths:**

* Scheduling DGNN snapshot groups from different time windows using Integer Linear Programming (ILP) or a greedy algorithm is an interesting idea. This approach effectively solve the load balance and inter-GPU communication problem, making it an intuitive and efficient solution.
* The paper has good coherence, and is well-structured.
* The paper is also very clear with thorough experiments and analysis.

**Weaknesses:**

* The primary optimization objective of DYGNEX is to reduce the total training time per epoch to a minimum. However, the training time is influenced by both the number of GPUs in use and the size of the workload. Consequently, a key consideration is the construction of the load from snapshot groups across different time windows. It is evident that varying snapshot sizes and diverse time intervals can significantly affect the load. As such, the overall performance of DYGNEX could be influenced. Please elucidate the effects of snapshot configuration on performance and explain how these configurations are managed.
* The sparsity of dynamic graphs can have a considerable impact on load distribution. How does DYGNEX mitigate the overhead associated with synchronization, especially given the potential variance between spectral and spatial work? My second concern is how DYGNEX achieves a balance between different Graph Neural Network (GNN) architectures and scales effectively with various types of dynamic graphs.
* In Figure 3, only a comparison of accuracy rates on the arXiv dataset is presented. However, the arXiv dataset does not clearly highlight the advantages of the method, as the training time per epoch is only reduced by about 0.1 seconds. This indicates that there is no significant improvement in training efficiency. Could you provide the impact on accuracy for other datasets? Could provide a further data illustrating the changes in accuracy rates across other datasets, such as the Reddit dataset?
* The scalability experiment in section 5.4 is somewhat misleading. At line 310, the author mentions that the experimental cluster consists of only 4 A100 GPUs. How were test data from other clusters obtained? Moreover, the paper does not clarify whether there is a linear relationship between throughput and the number of nodes, nor does it address whether using linear regression for prediction holds any practical value.
* There is no mention of the limitations of DYGNEX. Please include a subsection to address this point.

**Questions:**

* Please see the weakness section.

**Details Of Ethics Concerns:**

None.

---

> ### Author Response · Authors · 2024-11-22
>
> Thank you for your valuable review! Your insights have been instrumental in improving the quality of our paper, and we have uploaded a revised version based on your feedback. The revised sections have been highlighted in brickred for clarity.
>
> ### W1. Effects of snapshot configuration on performance and how these configurations are managed
>
> DyGNeX only needs to ensure that a single group can fit into the memory of one GPU. For the artificially created Arxiv, Products, and Reddit datasets, we followed the same approach as BLAD, sampling 30 snapshots from a static graph dataset, with the window size during training uniformly set to 4. For the Stackoverflow dataset, which spans 2774 days, we grouped the data from every 55 days into a single snapshot, with the training window size also set to 4.
>
> Snapshot configuration can impact the performance of DyGNeX. Our experiments on four datasets demonstrate the impact of snapshot size. We observed that DyGNeX achieves significant improvements across datasets with varying snapshot sizes. Notably, DyGNeX exhibits a greater advantage on datasets with larger snapshot sizes, such as Reddit. In such scenarios, the time imbalance effect caused by the differences among graphs is more pronounced. Time intervals affect the number of snapshot groups in a dataset, while the number of GPUs influences the number of snapshot groups trained per iteration. We can discuss the impact of these factors using the ratio $ \frac{\text{group count}}{\text{GPU count}} $. The results are presented in Section 5.4. When the $ \frac{\text{group count}}{\text{GPU count}} $ ratio is high—corresponding to scenarios with fewer GPUs in Figure 5—DyGNeX has sufficient flexibility in group scheduling, achieving nearly linear scalability. As the $ \frac{\text{group count}}{\text{GPU count}} $ ratio decreases, DyGNeX's scheduling flexibility becomes constrained, limiting the optimization potential. In this case, throughput experiences a certain degree of decline but still shows significant improvement compared to PSG.
>
> ---
>
> ### W2. How sparsity affect performance & Addressing synchronization overhead & Scalability across GNN architectures and dynamic graph types
>
> Admittedly, sparsity can affect the training time of snapshot groups; however, there are many factors influencing the training time of snapshot groups, and DyGNeX does not schedule groups based on these factors. **During the profiling phase, DyGNeX obtains the training time for each snapshot group and schedules tasks based on this training time, rather than the properties of the graph itself (e.g., edges, nodes, sparsity).**
>
> Meanwhile, synchronization occurs only during the gradient AllReduce phase, introducing no additional overhead.  Could you kindly confirm if this is the synchronization you are referring to?
>
> Similarly, DyGNeX does not impose any specific requirements on DGNN models or DTDGs; it simply needs to determine the training time for each snapshot group under the given DGNN model. As a result, DyGNeX is capable of extending to all DTDG datasets and DGNN models.
>
> ---
>
> ### W3. Accuracy
>
> We  the test accuracy on the Products and Reddit datasets in Figure 3.  This indicates that DyGNeX’s approach does not have a significant impact on accuracy.
>
> ---
>
> ### W4. Misleading in scalability experiment
>
> In Appendix H, we describe how the test data was obtained. Using Dygraph (GRADES-NDA'22), we generated 10,000 snapshots and profiled a subset of these snapshots on A100 GPUs. Through profiling, we observed that training time can be fitted using the equation below. Using another set of data for extrapolation, we found that the prediction error was less than 5%, as shown in Figure 14. We consider this prediction error to be within an acceptable range. Using this linear regression model, we predicted the training times for all 10,000 snapshots, which provided the test data used in our experiments.
>
> $t_{\text{snapshot}} = \alpha_1 \cdot N_{\text{node}} + \alpha_2 \cdot N_{\text{edge}} + \alpha_3$
>
>
> ---
>
> ### W5. Limitations
>
> We have added a section titled “Limitations” in the main text to discuss the limitations of DyGNeX and potential optimization directions.
>
> **Large Snapshot Group** In DyGNeX, each snapshot group must fit on a single GPU for training. While current GPU memory (e.g., 80GB, 40GB) suffices for most datasets, larger datasets may exceed this limit, making DyGNeX infeasible. A potential solution is to partition the graph and use full-neighbor sampling on target nodes, enabling training until all nodes are processed. The trade-offs between the overhead of partitioning and sampling and the advantages of DyGNeX over vertex-based methods (e.g., DGC) merit further exploration.
>
> **Limited Number of Snapshot Groups** The benefits of DyGNeX depend on the flexible combination of snapshot groups for load balancing. With very few groups, the limited combination space reduces potential gains.
>
> We hope our answers will address your concerns.

---

> ### Author Response · Authors · 2024-11-27
>
> Dear Reviewers,
>
> We would like to extend our heartfelt gratitude for the time and effort you have dedicated to reviewing our manuscript and providing us with such valuable suggestions. As the author-reviewer discussion phase is approaching its conclusion, we deem it necessary to confirm whether our responses have successfully tackled your concerns.
>
> A few days ago, we furnished detailed responses to address your concerns. We earnestly hope that these responses have satisfactorily resolved the issues you raised. In the event that you need further clarification or happen to have any additional concerns, please feel free to reach out to us at any time. We are wholeheartedly prepared to continue our communication with you and are committed to ensuring that all your queries are properly addressed.

---

### Official Review · Reviewer_3qzp · 2024-11-05

**Soundness:** 4
**Presentation:** 3
**Contribution:** 3
**Rating:** 6
**Confidence:** 2

**Summary:**

This work describes a method, DYGNEX, to improve the load balance and reduce the communication overhead of dynamic GNN training by distributing group-based snapshots of tasks between a collection of compute resources. DYGENEX first measures the training time required by each task then dynamically combines tasks for different time windows to improve load balance across all compute resources. The task groupings are computing using two different strategies, the first utilizes integer linear programming solvers to find a near-optimal assignment, at the expense of longer latency to find a solution, and a greedy method that finds a reasonable solution but not optimal considerably faster. Support for the performance of DYGNEX compared to other methods is provided in the evaluation section for different architectures, datasets, and data-partitioning strategies. The ILP solver reduces the training time per epoch considerably across different architectures and datasets while the greedy method outperforms competing methods but still underperforms the ILP approach. The merits of the greedy approach are analyzed using a distributed simulation for a larger cluster of devices where the time to compute the solution in the ILP approach becomes prohibitively expensive.

**Strengths:**

- The proposed method outperforms comparable approaches for a range of different architectures, datasets, and data-partitioning approaches. The evaluation of the methods across all these hyperparameters is sufficient to draw out the novelty and innovations of the proposed method and the additional notes in the Appendix help to clarify some of the details missing in the main body of the text.
- The tradeoff between the ILP and greedy methods is an interesting comparison and highlights the overhead associated with the near-optimal ILP solution and a greedy alternative. Table 3 is especially interesting for comparing the proposed method to previous approaches and between the 2 variations described in the text.
- The main text, along with the Appendix, is generally well-written and provides enough details to follow the implementation of the analysis of DYGNEX.

**Weaknesses:**

- While the gains compared to ESDG and BLAD are substantial the improvement compared with PSG is marginal in many cases, especially in comparison with the greedy variation.
- The ILP solver is computationally expensive but it is unclear if loosening the optimality constraint would allow the solver to work faster and achieve results on par or better than the greedy variation. It may be beneficial to provide a cost analysis of the ILP solver vs the greedy variation as the optimality constraint is increased from 1% to 10%.
- The use of a simulator instead of the actual runs is not ideal but does provide sufficient evidence that the ILP solve time and complexity would grow prohibitively with the number of compute resources.

**Questions:**

- How does varying the number of snapshots impact the training time per epoch and the task scheduling overhead? It seems to be fixed at 30 in section 4.3.
- How quickly does the dynamic connectivity of the graph change between epochs? I wonder if starting from a solution in a previous epoch would be informative to seed the solution in the current epoch and lower the time to compute a solution.

---

> ### Author Response · Authors · 2024-11-18
>
> Thank you for your valuable review! Your insights have been instrumental in improving the quality of our paper, and we have uploaded a revised version based on your feedback. The revised sections have been highlighted in brickred for clarity.
>
> ### W1. Improvement compared with PSG
>
> PSG is a baseline proposed in DyGNeX, and comparing it with PSG directly highlights the benefits brought by improved load balance. DyGNeX-G achieves an average improvement of 12.51% compared to PSG, while DyGNeX-L achieves an average improvement of 17.45%. We believe these improvement rates are considerable.
>
> ### W2. Cost analysis of the ILP solver vs the greedy variation
>
> We find your proposed viewpoint very meaningful. We present the solving times of DyGNeX-G and DyGNeX-L under different numbers of snapshots and gap constraints in the main text Table 5 and Table 6 (Word limit in comments). Based on the results in the tables, we observe that even with a relaxed 10% constraint, DyGNeX-L cannot match the speed of DyGNeX-G. From our experience, the performance difference between DyGNeX-G and DyGNeX-L (with a 2% constraint) is generally around 5%. Therefore, we believe that setting DyGNeX-L’s constraint to 2% or 3% compared to DyGNeX-G provides stable performance improvements. However, considering the computational cost of the ILP solver, we use DyGNeX-L for solving when the number of snapshots is less than 100 and DyGNeX-G when the number exceeds 100.
>
> ### W3. The use of a simulator instead of the actual runs is not ideal
>
> Except for Section 5.4, all of our experiments were **actually run on A100 GPUs**. In Section 5.4, due to hardware limitations, we used a simulator to conduct throughput simulations. Meanwhile, we explained the rationality of the simulation in Appendix H. In this section, we used the greedy algorithm for scheduling instead of the ILP method, because ILP was difficult to solve in this scenario, while the greedy algorithm could be quickly solved within 10 seconds, which we considered acceptable.
>
>
> ### Q1. How does varying the number of snapshots impact the training time per epoch and the task scheduling overhead?
>
> The number of snapshots affects the number of snapshot groups. Since DyGNeX leverages the lack of temporal dependency between snapshot groups to enable **data parallel training**, the impact of the number of snapshots on training time per epoch is consistent with the effect of increasing the number of training batches in data parallel training. The impact on task scheduling overhead is shown in Table 5 and Table 6. When the number of snapshots exceeds 100, DyGNeX-L experiences significant task scheduling overhead, and when the number of snapshots exceeds 10,000, DyGNeX-G faces noticeable task scheduling overhead.
>
> ### Q2. How quickly does the dynamic connectivity of the graph change between epochs?
>
> In DTDG training, the dataset is represented as  $G = (G_1, G_2, \dots, G_T)$, where the dynamic connectivity of the graph can change between iterations within an epoch but **remains unchanged between epochs**. Each epoch trains on the full sequence of graph changes, $G = (G_1, G_2, \dots, G_T)$. Therefore, **profiling and solving need to be done only once**, regardless of the number of epochs, and the resulting strategy can be applied to the entire training process.
>
> We hope our answers will address your concerns.

---

> > ### Comment · Reviewer_3qzp · 2024-11-26
> >
> > I want to thank the authors for their detailed response to my review and for providing an updated paper version. Based on their response I will maintain my current rating.

---

### Official Review · Reviewer_VVtG · 2024-11-07

**Soundness:** 3
**Presentation:** 3
**Contribution:** 2
**Rating:** 6
**Confidence:** 3

**Summary:**

DyGNeX is a distributed training system for dynamic GNN. It achieves minimal communication and balanced workload.
- To achieve (1), data parallelism is adopted so that only model weights/gradients need to be synchrnoized
- To achieve (2), workload distribution is formulated and is optimized via ILP.

**Strengths:**

- This paper studies an important research topic
- The system-level performance is promising.
- Experimental results show that DyGNeX boosts the efficiency of 4 popular dynamic GNN architecture.

**Weaknesses:**

- Insufficient experiments.

**Questions:**

- My major concern is the test accuracy of the proposed method. Only one dataset is used for evaluating accuracy. It's not clear whether the proposed method hurts model performance significantly.
- Memory consumption of DyGNeX is not reported. However, this overhead might be significant as all node features are stored in each GPU (line 104).

---

> ### Author Response · Authors · 2024-11-18
>
> Thank you for your valuable review! Your insights have been instrumental in improving the quality of our paper, and we have uploaded a revised version based on your feedback. The revised sections have been highlighted in brickred for clarity.
>
> ### Q1. Test accuracy
>
> We present the test accuracy on the Products and Reddit datasets in Figure 3.It can be observed that in the vast majority of scenarios, there is no difference in accuracy between DyGNeX and PSG. This indicates that DyGNeX’s approach does not have a significant impact on training quality.
>
> ### Q2. Memory consumption
>
> We use `torch.cuda.max_memory_allocated()` to analyze the memory consumption during training. As shown in the table below, we observed that DyGNeX consistently requires less GPU memory compared to BLAD across all datasets and models. The results demonstrate that DyGNeX has lower memory requirements compared to BLAD. In DyGNeX, all experiments can be conducted on consumer-grade GPUs, such as the NVIDIA RTX 2080 Ti or RTX 3090. Due to the single-GPU dual-process pipeline handling groups, BLAD has introduced additional memory consumption caused by model replicas and so on.
>
> #### Memory Consumption Comparison between DyGNeX and BLAD (in GB)
>
> | **Model**        | **Method** | **Arxiv** | **Products** | **Reddit** | **Stackoverflow** |
> |-------------------|------------|-----------|--------------|------------|-------------------|
> | **EvolveGCN**     | BLAD       | 1.14      | 3.52         | 9.11       | 0.67              |
> |                   | DyGNeX     | 0.79      | 3.10         | 8.74       | 0.63              |
> | **WDGCN**         | BLAD       | 4.29      | 14.2         | 36.07      | 5.42              |
> |                   | DyGNeX     | 0.76      | 3.09         | 8.68       | 0.63              |
> | **TGCN**          | BLAD       | 1.13      | 3.47         | 9.02       | 0.70              |
> |                   | DyGNeX     | 0.77      | 2.93         | 8.68       | 0.63              |
> | **GAT-LSTM**      | BLAD       | N/A       | N/A          | N/A        | N/A               |
> |                   | DyGNeX     | 0.79      | 2.93         | 8.68       | 0.63              |
>
> We hope our answers will address your concerns.

---

### Note · Authors · 2025-01-18

I have read and agree with the venue's withdrawal policy on behalf of myself and my co-authors.